# Physics Informed Machine Learning of SPH: Machine Learning Lagrangian Turbulence

## Abstract

Smoothed particle hydrodynamics (SPH) is a mesh-free Lagrangian method for obtaining approximate numerical solutions of the equations of fluid dynamics, which has been widely applied to weakly- and strongly compressible turbulence in astrophysics and engineering applications. We present a learn-able hierarchy of parameterized and "physics-explainable" SPH informed fluid simulators using both physics based parameters and Neural Networks as universal function approximators. Our learning algorithm develops a mixed mode approach, mixing forward and reverse mode automatic differentiation with forward and adjoint based sensitivity analyses to efficiently perform gradient based optimization. We show that our physics informed learning method is capable of: (a) solving inverse problems over the physically interpretable parameter space, as well as over the space of Neural Network parameters; (b) learning Lagrangian statistics of turbulence; (c) combining Lagrangian trajectory based, probabilistic, and Eulerian field based loss functions; and (d) extrapolating beyond training sets into more complex regimes of interest. Furthermore, our hierarchy of models gradually introduces more physical structure, which we show improves interpretability, generalizability (over larger ranges of time scales and Reynolds numbers), preservation of physical symmetries, and requires less training data.

## 1 Introduction

**Why turbulence? Why Lagrangian?** Understanding turbulent flows is crucial for many engineering and scientific fields, and remains a great unresolved challenge of classical physics (Frisch, 1995). Many computational fluid dynamics (CFD) approaches (Harlow, 2004) are being explored in this direction by approximating the Navier-Stokes (NS) equations which are well established in the field as explaining the "ground truth". One such CFD approach is the mesh-free Smoothed Particle Hydrodynamics (SPH) method (Gingold & Monaghan, 1977; Monaghan, 1992; 2012), which has been widely applied to weakly- and strongly compressible turbulence in astrophysics and many engineering applications (Shadloo et al., 2016). It is one of a very few approaches based on a Lagrangian construction: fluid quantities follow the flow using particles as opposed to the Eulerian approach which computes flow quantities at fixed locations by using a computational mesh. This mesh-free, Lagrangian approximation of NS is appealing because it naturally unmasks correlations at the resolved scale from sweeping by larger scale eddies (Kraichnan, 1964; 1965). Developing approximation-optimal SPH models, and then simulators, for turbulent flows is an ongoing area of research (Lind et al., 2020), to which this paper contributes. Specifically, we show how modern tools of machine learning and applied mathematics, such as deep neural networks (NNs), automatic differentiation (AD), and sensitivity analysis (SA), can be utilized and developed to design better SPH models.

**Physics Informed Machine Learning: What and Why?** Numerical simulators, including classical CFD simulators, have recently been blended with modern machine learning tools (King et al., 2018; Schenck & Fox, 2018; Mohan & Gaitonde, 2018; Mohan et al., 2020a; Maulik et al., 2020; Ummenhofer et al., 2020; Mohan et al., 2020b; Tian et al., 2021) out of which a promising field, coined Physics Informed Machine Learning (PIML), is emerging (and being re-discovered (Lagaris et al., 1998)). As set, some eight years ago at the first LANL workshop with this name (CNLS at LANL, 2016, 2018, 2020), PIML was meant to pivot the mixed community of machine learning researchers on one hand and scientists and engineers on the other, to discover physical phenomena/models from data. Today, PIML researchers are focused on incorporating known physical structures (such as conservation laws), physical hypothesis and physical equations (and/or their numerical

schemes/algorithms) into modern machine learning tools. This development is exciting for physicists, as it integrates centuries of scientific discovery with modern machine learning – boosting new ideas and resurrecting interesting but forgotten hypotheses from the past. The overall goal of PIML is two-fold (King et al., 2018): learning new physics from data (experimental or computational) and improving/developing machine learning algorithms by incorporating ideas from physics (especially in what concerns explainable, interpretable, and generalizable models).

**Our Contribution: PIML for SPH.** We develop a learn-able hierarchy of parameterized "physics-explainable" SPH based fluid simulators that are trained and analyzed on SPH flow data. We train these models by mixing automatic differentiation (both forward and reverse mode) with forward and adjoint based sensitivity analyses. Utilizing these models and methods, we show that adding physical structure improves interpretability, generalizability, and requires less training data compared to less informed models (such as Neural ODE). Additionally, we show the principal ability of these parameterized SPH simulators to solve inverse problems, and be fitted to the ground truth flow data using a combination of field based and statistical based loss functions. Furthermore, we show that the learned models are able to reconstruct the Lagrangian statistics of the flow, and can be used to learn unknown, or missing, functions embedded within the SPH models by using Neural Networks as function approximators (Hornik, 1989).

**Why PIML challenges are of interest to a broader Machine Learning Community?** Machine Learning has made tremendous progress in recent years by developing a plethora of sound practical methods, algorithms and software which allow one to work with all kinds of data in an application agnostic way. Today we can solve many challenging fitting and prediction problems of interpolation type, which were unthinkable even five years ago. However, there are still challenges that remain – notably one related to extrapolation, and thus generalizability, into regimes where data is not sufficient, if available at all. One way to approach these challenges is to substitute lack of data with application-specific modeling – in our case Lagrangian Modeling of Turbulent Flows. The emerging PIML community this research belongs to is focusing exactly on this. We are making progress, but also need help from a broader machine learning community on multiple issues, in particular these highlighted in this paper: How to embed physics-based constrains/symmetries/dependencies into ML frameworks? How does embedding physics into ML schemes contribute (simplify or complicate) to roughness of the training landscape? How to balance Bayesian and deterministic approaches, e.g. selecting loss functions, in the intrinsically stochastic extrapolation problems, like turbulence? Finally, how to balance physics explanations (by tuning physically meaningful parameters, e.g. with sensitivity approaches) with predictive capabilities of machine learning (by training NN parameters, introduced for degrees of freedom which we do not need or do not know how to interpret)?

**Outline.** The overall outline of the remainder of the manuscript is a follows. First, in section 2, an overview of related studies is given that draws connections of this manuscript to the broader field of PIML. Next, in section 3, and Appendix A, a formulation of SPH is presented, which provides the basis for our work on learning Lagrangian models of turbulence using parameterized and learn-able SPH simulators. In section 4 and Appendix B, a detailed description of the learning algorithms, loss functions, and hierarchy of models are suggested, which provides the necessary tools to learn parameters of the SPH model. Then, in section 5 and Appendix C, we analyze this hierarchy of models and address how adding more of the known physical (SPH) structure affects the ability of the models to generalize to longer times and larger Reynolds numbers ($Re$). Finally, in section 6, we draw conclusions on the progress made so far along with providing a discussion of future work.

## 2 RELATED WORK

Early works of incorporating scientific domain knowledge (in the form of differential equations) within deep learning algorithms dates back to the 1990s (Lagaris et al., 1998). However, in the context of modern deep learning, interest in this area has been revived (Raissi et al., 2017; Chen et al., 2019; Rackauckas et al., 2020; Ladický et al., 2015), due in part to the increased computational power afforded by parallelism across both CPUs and GPUs, along with notable achievements across disciplines (such as scientific applications (Zhai et al., 2020), data-compression algorithms (Wang et al., 2016), computer vision (Serre, 2019), natural language processing (Li Deng, 2018), etc. The main computational utilities and strategies of physics informed learning includes using physical

structure, NNs as function approximators (Hornik, 1989), automatic differentiation (Bücker, 2006), and optimization tools to minimize an objective (loss) function.

There are several approaches that can be taken to develop a PIML algorithm; (1) using soft constraints by adding physical structure, or symmetries into the loss function (as was done in (Raissi et al., 2017) where Neural Networks are used to discover and solve parameterized PDEs from data), (2) enforce physical constraints directly into the NN architecture (as (Mohan et al., 2020a) enforced the incompressiblility constraint into a Convolutional NN), (3) utilize NN's as function approximators along with physical parameters/structure embedded within differentiable numerical simulators (as was done in (Chen et al., 2019; Rackauckas et al., 2020; Tian et al., 2021)), (4) mixing differentiable programming with SA along with physical structure and NNs as was done by (Ma et al., 2021; Chen et al., 2019) (which from a technical perspective is most related to our work). The broader focus of this work is on (4) as *an approach to build parameterized Lagrangian models as candidates for SPH models of turbulence. However, in this manuscript, we combine these scientific disciplines to not only provide a tool for approaching the discovery of optimal Lagrangian models for turbulence, but also to explore the effects of adding physical structure into ML algorithms on extrapolation and generalizability.*

In recent years, there have been several works that have integrated ML and DL techniques for Lagrangian flows. The pioneering work by (Ladický et al., 2015) used SPH related models and regression forests along with physics informed feature vectors, demonstrating that flow representations can be learned with data-driven techniques. In (de Anda-Suárez et al., 2018), evolutionary algorithms are applied for optimization of parameters in SPH flows. Further works began utilizing differentiable programming techniques, such as in (Schenck & Fox, 2018) for robotic control for pouring applications and in (Ummenhofer et al., 2020) where a continuous convolutional neural network operation is developed on unordered particle data for learning and simulating SPH. (Tian et al., 2021) learns reduced models describing the Lagrangian dynamics of the velocity gradient tensor from Direct Numerical Simulation data. Other works have used experimental data from Particle Image Velocimetry and NNs to track particles embedding in a flow (Rabault et al., 2017; Lee et al., 2017; Cai et al., 2019; Stulov & Chertkov, 2021).

## 3    SMOOTHED PARTICLE HYDRODYNAMICS

One of the most prominent particle-based Lagrangian methods for obtaining approximate numerical solutions of the equations of fluid dynamics is Smoothed Particle Hydrodynamics (SPH) (Monaghan, 2005). Originally introduced independently by (Lucy, 1977) and (Gingold & Monaghan, 1977) for astrophysical flows, however, over the following decades, SPH has found a much wider range of applications including computer graphics, free-surface flows, fluid-structure interaction, bioengineering, compressible flows, galaxies' formation and collapse, high velocity impacts, geological flows, magnetohydrodynamics, and turbulence (Lind et al., 2020; Shadloo et al., 2016). Below, we give a brief formulation of SPH and in subsection 4.1 we hard code SPH structure into a hierarchy of "physics-explainable" models.

### 3.1    APPROXIMATION OF EQUATIONS OF MOTION

Essentially, SPH is a discrete approximation to a continuous flow field by using a series of discrete particles as interpolation points (using an integral interpolation with smoothing kernel $W$). Using the SPH formalism (see Appendix A for more details), the partial differential equations (PDEs) of fluid dynamics can be approximated by a system of ordinary differential equations (ODEs) for each particle (indexed by $i$)

$$\frac{d\boldsymbol{r}_i}{dt} = \boldsymbol{v}_i, \quad \frac{d\boldsymbol{v}_i}{dt} = -\sum_{j\neq i}^{N} m_j \left( \frac{P_j}{\rho_j^2} + \frac{P_i}{\rho_i^2} + \Pi_{ij} \right) \nabla_i W_{ij} + \boldsymbol{f}_{ext}, \quad \forall i \in \{1, 2, ...N\}. \quad (1)$$

See Appendix A for a more detailed derivation, as well as description of how particle density $\rho_i$ and pressure $P_i$ are computed as in Monaghan (2012). In this manuscript, deterministic and stochastic external forcing $\boldsymbol{f}_{ext}$ was explored (see Appendix A for more details), which provides the energy injection mechanism. We also utilize the popular formulation of an artificial viscosity $\Pi_{ij}$, which approximates in aggregate contributions from the bulk and shear viscosity ($\alpha$), a Nueman-Richtmyer

viscosity for handling shocks ($\beta$) Monaghan (2012); Morris et al. (1997), as well as the effective, eddy viscosity effect of turbulent advection from the under-resolved scales, i.e. scales smaller than the mean-particle distance.

## 3.2 EXAMPLE FLOW: TRAINING SET

Fig. (1) shows consecutive snapshots of an exemplary multi-particle SPH flow in three dimensional space, where the coloration is added for visualization purposes. We use a standard set of parameters for weakly-compressible flows (see Cossins (2010)) $\alpha = 1.0$ (bulk-shear viscosity), $\beta = 2\alpha$ (Nueman-Richtmyer viscosity) , $c_s = 10$ (speed of sound), $\gamma = 7.0$ with energy injection rate $\theta = 0.5$, and deterministic external forcing,(Appendix A). The inverse problem we pose consists in, given a sequence of snapshots, to reconstruct as best as we can the underlying (SPH) model used to generate the data.

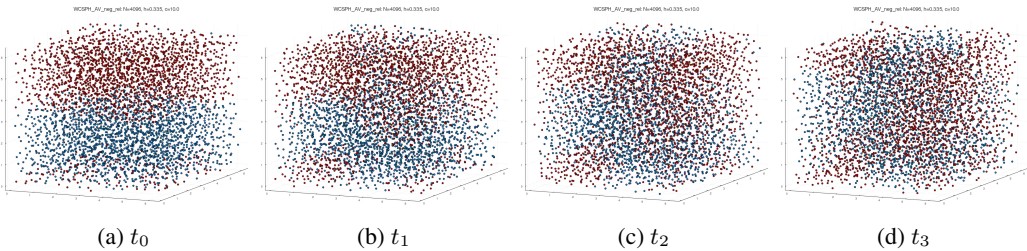

(a) $t_0$        (b) $t_1$        (c) $t_2$        (d) $t_3$

Figure 1: SPH particles advancing in time used in training data, where coloration is added for visualization purposes and $t_{i+1} - t_i \approx 100\Delta t$.

## 4 MIXED MODE PHYSICS INFORMED MACHINE LEARNING METHODS

This section gives a brief overview of our mixed mode physics informed learning algorithm (a first order gradient based optimization approach). This includes the hierarchy of parameterized SPH informed models, methods for computing the gradients, and formulating the loss functions. These methods are implemented using the open source software Julia (Bezanson et al., 2017), which is a fast and flexible dynamic language, appropriate for high performance scientific and numerical computing, along with a wide range of data science and machine learning packages such as Flux.jl (Innes et al., 2018), and AD packages Zygote.jl (Innes, 2018), ForwardDiff.jl (Revels et al., 2016). For a more detailed treatment on our learning algorithms (including derivations), and loss functions see Appendix B.

### 4.1 HIERARCHY OF MODELS

In this work, we hard code SPH structure into a hierarchy of parameterized models that includes physics based and Neural Network based parameters. Multilayer Perceptrons (MLPs) are used as a universal function approximators (Hornik, 1989) embedded within an ODE structure describing the Lagrangian flows. It was found through hyper-parameter tuning that 2 hidden layers were sufficient for each model using a NN.

• **NODE:** In this least informed model that we explore, the entire acceleration term, that is the entire right hand side of Eq. (1) with the exception of $\boldsymbol{f}_{ext}$, is approximated by an MLP, which is most related to the work done by (Chen et al., 2019). We make an additional modification by considering the interaction of particles to be within a local cloud (using the cell linked list algorithm (Domínguez et al., 2011)). Note here that no pairwise interaction between particles is assumed. We assume that velocities, $\boldsymbol{v}_i(t)$, and coordinates, $\boldsymbol{r}_i(t)$, of $N$ particles evolve in time according to

$$\frac{d\boldsymbol{r}_i}{dt} = \boldsymbol{v}_i, \quad \frac{d\boldsymbol{v}_i}{dt} = \mathbf{NN}_{\boldsymbol{\theta}}\left(\boldsymbol{r}_i - \boldsymbol{r}_j, \boldsymbol{v}_i - \boldsymbol{v}_j \big| \forall j : |\boldsymbol{r}_i - \boldsymbol{r}_j| \leq 2h\right) + \boldsymbol{f}_{ext} \quad \forall i = 1, \cdots, N \quad (2)$$

where $\mathbf{NN}_{\boldsymbol{\theta}} : \mathbb{R}^{2dm} \to \mathbb{R}^l \to \mathbb{R}^l \to \mathbb{R}^d$ ($d = 2, 3$ is the space dimension, $m$ is the fixed number of particles that are closest to the $i$-th particle in each cloud, and $l$ is the height, or number of

nodes, of the hidden layer). Although this $NN$ is approximating a function which is interpretable (acceleration), the individual parameters of the $NN$ are not.

• **NN summand:** To include more of the physical structure, represented via a sum over the $i$-th particle neighborhood in Eq. (1) (which is embedded in generating the "synthetic" ground truth data), we approximate the summand term from SPH using a NN. Now a pairwise interaction between particles is assumed. Here, Eq. (1) is modeled by the Lagrangian based ODE

$$\frac{d\boldsymbol{r}_i}{dt} = \boldsymbol{v}_i, \quad \frac{d\boldsymbol{v}_i}{dt} = \sum_j^N \boldsymbol{NN_\theta}\left(\boldsymbol{r}_i - \boldsymbol{r}_j, \boldsymbol{v}_i - \boldsymbol{v}_j\right) + \boldsymbol{f}_{ext} \quad \forall i = 1, \cdots, N \quad (3)$$

where $NN_{\boldsymbol{\theta}} : \mathbb{R}^{2d} \to \mathbb{R}^l \to \mathbb{R}^l \to \mathbb{R}^d$.

• **Rotationally Invariant NN**: In this formulation, built on the top of the NN summand, we use a neural network of the form $NN_{\boldsymbol{\theta}} : \mathbb{R}^4 \to \mathbb{R}^l \to \mathbb{R}^l \to \mathbb{R}$ to approximate the pair-wise part of the acceleration term in Eq. (1), where the rotational invariance is hard coded by construction (about a simple rotationally invariant basis expansion using the difference vector $(\boldsymbol{r}_i - \boldsymbol{r}_j)$), as follows

$$\frac{d\boldsymbol{r}_i}{dt} = \boldsymbol{v}_i, \quad \frac{d\boldsymbol{v}_i}{dt} = \sum_j^N NN_{\boldsymbol{\theta}}\left(\frac{P_i}{\rho_i^2}, \frac{P_j}{\rho_j^2}, (\boldsymbol{r}_i - \boldsymbol{r}_j) \cdot (\boldsymbol{v}_i - \boldsymbol{v}_j), ||\boldsymbol{r}_i - \boldsymbol{r}_j||_2\right)(\boldsymbol{r}_i - \boldsymbol{r}_j) + \boldsymbol{f}_{ext} \quad (4)$$

• **$\nabla P$- NN:** Next, we push the neural network deeper within the summand term and now explicitly include the $\Pi$-, i.e. artificial viscosity term, and use a $NN$ to approximate the pressure contribution (i.e $\nabla P$- term in SPH Eq. (1), see Appendix A for more details):

$$\frac{d\boldsymbol{r}_i}{dt} = \boldsymbol{v}_i, \quad \frac{d\boldsymbol{v}_i}{dt} = \sum_j^N (NN_{\boldsymbol{\theta}}\left(\boldsymbol{r}_i - \boldsymbol{r}_j\right) + \Pi_{ij})\nabla W_{ij} + \boldsymbol{f}_{ext} \quad \forall i = 1, \cdots, N \quad (5)$$

where $NN_{\boldsymbol{\theta}} : \mathbb{R}^d \to \mathbb{R}^l \to \mathbb{R}^l \to \mathbb{R}$.

• **EoS NN:** Embedding a $NN$ within an SPH simulator for learning the interpretable equation of state from flow data using a neural network ($Pnn_{\boldsymbol{\theta}}$) for approximating $P(\rho)$ in Eq. (1):

$$\frac{d\boldsymbol{r}_i}{dt} = \boldsymbol{v}_i, \quad \frac{d\boldsymbol{v}_i}{dt} = \sum_j \left(\frac{Pnn_{\boldsymbol{\theta}}(\rho_i)}{\rho_i^2} + \frac{Pnn_{\boldsymbol{\theta}}(\rho_j)}{\rho_j^2} + \Pi_{ij}\right)\nabla W_{ij} + \boldsymbol{f}_{ext} \quad \forall i = 1, \cdots, N \quad (6)$$

where $Pnn_{\boldsymbol{\theta}}(\rho) : \mathbb{R} \to \mathbb{R}^l \to \mathbb{R}^l \to \mathbb{R}$.

• **Fully Physics Informed:** In this formulation, the entire physical structure, that is known to generate the ground truth, is used, and the physically interpretable parameters $\alpha, \beta, \gamma, c$ are learned

$$\frac{d\boldsymbol{r}_i}{dt} = \boldsymbol{v}_i, \quad \frac{d\boldsymbol{v}_i}{dt} = \sum_j \left(\frac{P_i(c, \gamma)}{\rho_i^2} + \frac{P_j(c, \gamma)}{\rho_j^2} + \Pi_{ij}(c, \alpha, \beta)\right)\nabla W_{ij} + \boldsymbol{f}_{ext} \quad \forall i = 1, \cdots, N. \quad (7)$$

We note that as more of the physical structure is added into the learning algorithm, the learned models have more interpretability; the learned parameters are associated with actual physical quantities.

## 4.2 MIXING SENSITIVITY ANALYSIS AND AUTOMATIC DIFFERENTIATION

Sensitivity analysis (SA) is a classical technique found in many applications, such as gradient-based optimization, optimal control, parameter identification, model diagnostics, (see (Donello et al., 2020) and many historical references there in), which was also utilized recently to learn neural network parameters within ODEs (Chen et al., 2019; Rackauckas et al., 2020; Ma et al., 2021). In the context of this work, we use SA to compute gradients of our hierarchy of parameterized models. We mix SA with Automatic Differentiation (AD) (see (Ma et al., 2021; Bücker, 2006) and references there in): forward mode and reverse mode AD is applied to derivatives within the SA algorithm, where the method is chosen based on efficiency (depending on the dimension of the input and output space of the function being differentiated). The losses used in this work are defined in Eq. (4.4), but in general, we consider loss functions of the form, $L(\boldsymbol{X}, \boldsymbol{\theta}) = \int_0^{t_f} \Psi(\boldsymbol{X}, \boldsymbol{\theta}, t)dt$, where $\boldsymbol{X}$ and $\boldsymbol{\theta}$ are, respectively, the vector of the combined vector of the particles' coordinates and velocities and the vector of parameters introduced in the next subsection.

### 4.2.1 Forward and Adjoint based Methods

Let us, first, introduce some useful common notations for our SPH informed models, $\boldsymbol{X}_i = (\boldsymbol{r}_i, \boldsymbol{v}_i)^T$, $\boldsymbol{X} = \{\boldsymbol{X}_i | i = 1, ... N\}$, where, $\boldsymbol{r}_i = (x_i, y_i, z_i)$, and, $\boldsymbol{v}_i = (u_i, v_i, w_i)$, are the position and velocity of particle $i$ respectively. Also, $\boldsymbol{\theta} = [\theta^1, ..., \theta^p]^T$ where $p$ is the number of model parameters. Now, the SPH discretizations defined in Eq. (1) can be stated in the ODE form

$$\forall i: \quad d\boldsymbol{X}_i/dt = \boldsymbol{\mathcal{F}}_i(\boldsymbol{X}(t, \boldsymbol{\theta}), \boldsymbol{\theta}) = (\boldsymbol{v}_i, \ \boldsymbol{F}_i(\boldsymbol{X}, \boldsymbol{\theta}))^T \tag{8}$$

where $\boldsymbol{F}_i$ is the right hand side of Eq. (1). Forward and Adjoint based Sensitivity Analyses (FSA, ASA), analogous to forward and reverse mode AD respectively, are used to compute the gradient of the loss function (see Section 4.4), $\partial_{\boldsymbol{\theta}} L(\boldsymbol{X}, \boldsymbol{\theta}) = \int_0^{t_f} \partial_{\boldsymbol{X}} \Psi(\boldsymbol{X}, \boldsymbol{\theta}, t) d_{\boldsymbol{\theta}} \boldsymbol{X}(\boldsymbol{\theta}, t) + \partial_{\boldsymbol{\theta}} \Psi(\boldsymbol{X}, \boldsymbol{\theta}, t) dt$, (for derivations of FSA and ASA see Appendix B). FSA computes the sensitivities, $\boldsymbol{S}_i^{\alpha} = d\boldsymbol{X}_i/d\theta^{\alpha}$, by simultaneously integrating a system of ODEs:

$$\forall i, \ \forall \alpha = 1, \cdots, p: \quad d\boldsymbol{S}_i^{\alpha}/dt = (\partial \boldsymbol{\mathcal{F}}_i(\boldsymbol{X}(t), \boldsymbol{\theta})/\partial \boldsymbol{X}_i) \boldsymbol{S}_i^{\alpha} + \partial \boldsymbol{\mathcal{F}}_i(\boldsymbol{X}(t), \boldsymbol{\theta})/\partial \theta^{\alpha}, \tag{9}$$

which is known to be more efficient when the number of parameters, $p$, is not very large, $p \lesssim \mathcal{O}(100)$. Once $\boldsymbol{S}_i^{\alpha}$ is known, the gradient is computed directly from the formula for $\partial_{\boldsymbol{\theta}} L$ as above. The ASA avoids needing to compute $d_{\boldsymbol{\theta}} \boldsymbol{X}$ by instead numerically solving a system of equations for the adjoint equation backwards in time, according to subsubsection B.0.2. Once $\boldsymbol{\lambda}$ is found, the gradient of the loss function is found according to, $\partial_{\boldsymbol{\theta}} L = - \int_0^{t_f} \sum_i \boldsymbol{\lambda}_i (\partial \mathcal{F}_i/\partial \boldsymbol{\theta}) dt$. It is well known that the ASA is more efficient when $p \gtrsim \mathcal{O}(100)$ (see (Ma et al., 2021)), because solving the adjoint equation is independent of $p$ but requires more memory to store forward the solution then to integrate $\boldsymbol{\lambda}^T$ backwards in time.

### 4.3 Mixed Mode AD

In both FSA and ASA described above, the gradient of $\boldsymbol{\mathcal{F}}_i$ with respect to the parameters, $\partial \boldsymbol{\mathcal{F}}_i(\boldsymbol{X}(\tau), \boldsymbol{\theta})/\partial \boldsymbol{\theta}$, and the Jacobian matrix, $\{\partial \boldsymbol{\mathcal{F}}_i(\boldsymbol{X}(\tau), \boldsymbol{\theta})/\partial \boldsymbol{X}_j | \forall i, j\}$, need to be computed. In this manuscript, we accomplish this with a mixed mode approach, i.e. mixing forward and reverse mode AD, where the choice is based on efficiency. Depending on the model used, many of the functions to be differentiated have varying input and output dimensions. For example, when computing $\partial \boldsymbol{\mathcal{F}}_i(\boldsymbol{X}(\tau), \boldsymbol{\theta})/\partial \theta^{\alpha}$, with AD where, $\boldsymbol{\mathcal{F}}_i(\boldsymbol{\theta}) : \mathbb{R}^p \to \mathbb{R}^{2d}$, if $p >> 2d$, then reverse mode AD is more efficient than forward mode (Bücker, 2006).

### 4.4 Loss functions

In this section, we construct three different loss functions: trajectory based (Lagrangian), field based Eulerian, and Lagrangian statistics based, described in the following three subsections. Since our overall goal involves learning SPH models for turbulence applications, it is the underlying statistical features and large scale field structures we want our models to learn and generalize with. This is discussed further in section 5, where we compare the hierarchy of models that was constructed in subsection 4.1.

### 4.4.1 Trajectory Based Loss Function

A simple loss function to consider is the Mean Squared Error ($MSE$) of the difference in the Lagrangian particles positions and velocities, as they evolve in time, $L_{tr}(\boldsymbol{\theta}) = MSE(\boldsymbol{X}, \hat{\boldsymbol{X}}(\boldsymbol{\theta})) = \|\boldsymbol{X} - \hat{\boldsymbol{X}}(\boldsymbol{\theta})\|^2/N$, where $\boldsymbol{X}$ and $\hat{\boldsymbol{X}}$ are the particle states – the ground truth and the predicted, respectively. Minimizing this loss function will result in discovering optimal parameters such that the predicted trajectories gives the best possible match (within the model) for each of the particles. Notice that this "perfect" matching of the multi-particle state is not really appropriate for tracking temporal correlations in turbulence on the time scales longer than the turn-over time of the resolved eddy (estimated, roughly, as the time needed for a pair of initially neighboring particles to separate on a distance comparable to their initial separation). This is due to the fact that turbulence is intrinsically chaotic, therefore resulting in a strong sensitivity of the state (Lagrangian particle positions and velocities) on its initial conditions. Therefore, we expect the trajectory based loss method to over-fit when the observation (tracking) time is sufficiently long.

### 4.4.2 FIELD BASED LOSS FUNCTION

The field based loss function tries to minimize the difference between the large scale structures found in the Eulerian velocity fields, $L_f(\boldsymbol{\theta}) = MSE(\boldsymbol{V}^f, \hat{\boldsymbol{V}}^f) = \|\boldsymbol{V}^f - \hat{\boldsymbol{V}}^f\|^2/N_f$, where $\boldsymbol{V}_i^f = \sum_{j=1}^{N_f}(m_j/\rho_j)\boldsymbol{v}_j W_{ij}(\|\boldsymbol{r}_i^f - \boldsymbol{r}_j\|, h)$ uses the same SPH smoothing approximation to interpolate the particle velocity onto a predefined mesh $\boldsymbol{r}^f$ (with $N_f$ grid points). Let us remind that SPH is, by itself, an approximation for the velocity field, therefore providing a strong additional motivation for using the field based loss function.

### 4.4.3 LAGRANGIAN STATISTICS BASED LOSS FUNCTION

In order to approach the statistical nature of turbulent flows, one can use well established statistical tools/objects, such as single particle statistics. In this direction, consider the time integrated Kullback–Leibler divergence (KL) as a loss function

$$L_{kl}(\boldsymbol{\theta}) = \int_{t=0}^{t_f}\int_{-\infty}^{\infty} G_{gt}(t, \boldsymbol{z}_{gt}(t), \boldsymbol{x}) \log\left(\frac{G_{gt}(t, \boldsymbol{z}_{gt}(t), \boldsymbol{x})}{G_{pred}(t, \boldsymbol{z}_{pr}(\boldsymbol{\theta}, t), \boldsymbol{x})}\right) d\boldsymbol{x}dt,$$

where $\boldsymbol{z}_{gt}(t)$, and $\boldsymbol{z}_{pr}(\boldsymbol{\theta}, t)$ represent single particle statistical objects over time of the ground truth and predicted data, respectively. For example, we can use the velocity increment, $\boldsymbol{z}^i(t) = (\delta u_i, \delta v_i, \delta w_i)$, where $\delta u_i(t) = u_i(t) - u_i(0)$ and $\boldsymbol{z}$ ranges over all particles. Here $G(t, \boldsymbol{z}(t), x)$ is a continuous probability distribution (in $x$) constructed from data $\boldsymbol{z}(t)$ using Kernel Density Estimation (KDE), to obtain smooth and differentiable distributions from data, as discussed in (Chen, 2017)), that is $G(\tau, \boldsymbol{z}, \boldsymbol{x}) = (Nh)^{-1}\sum_{i=1}^{N} K\left((z_i - \boldsymbol{x})/h\right)$, where $K$ is the smoothing kernel (chosen to be the normalized Gaussian in this work). (See Appendix B.1 for details.) In 3D experiments (discussed in the next section), we use $L_{kl} + L_f$ as a combination of statistical and field based loss functions, with the expectation that the gradient descent (in the process of training) will drive different parameters, responsible for large and small structures, to their optimal values in unison.

## 5 RESULTS: EVALUATING AND TESTING THE HIERARCHY OF MODELS

In this section, we show the ability of the parameterized SPH simulators to; solve inverse problems, fit underlying flow data using a combination of field based and statistical based loss functions, learn unknown functions embedded within SPH based models, generalize to flows not seen in training and compare their ability to conserve linear and angular momentum. In subsection 5.1 we illustrate the ability of different models in the hierarchy to learn the parameters to approximate the (underlying) distribution of velocity increments. Then, in section 5.2 we focus on comparing the generalizability of different models in the hierarchy. We show that the more physical structure is hard-coded into the model, the better is the generalizability ( i.e. extrapolation over larger time scales and Reynolds numbers) and the better it conserves linear and angular momentum (preserves transnational and rotational symmetries respectively). We also notice that another important advantage of the physics informed models is that they require less training data (as seen in Figure 9).

### 5.1 SOLVING INVERSE PROBLEMS

Each parameterized SPH model within the hierarchy (see subsection 4.1) is trained under equivalent conditions: (a) on the same SPH samples (see subsection 3.2) of fixed temporal duration (which we choose to be equal to the time scale required for a pair of neighboring particles to separate by the distance which is in average factor $O(1)$ larger than the pair's initial separation and henceforth denoted $t_\lambda$); (b) with the same loss function; and (c) with a deterministic forcing $\boldsymbol{f}_{ext}$ (subsection A.3) with constant rate of energy injection.

In Figure 2 and Figure 6, we see that the physics informed parameterized SPH simulator is learnable; the physical parameters $\alpha, \beta, c, \gamma$ are learned over the physically interpretable parameter space. Figure 3 illustrates the ability of the method applied to the $EoSNN$ model to learn (approximate) physically interpretable functions ($P(\rho)$) using NNs embedded within an SPH model. Figure 4 shows that each model is capable of learning the parameters so that the underlying velocity increment distribution (a single particle Lagrangian statistic) is approximated, i.e learning Lagrangian statistics.

These figures, show that over a mixture of loss functions (subsection 4.4), applying the methods described in subsubsection 4.2.1, the SPH informed models (subsection 4.1) are learn-able.

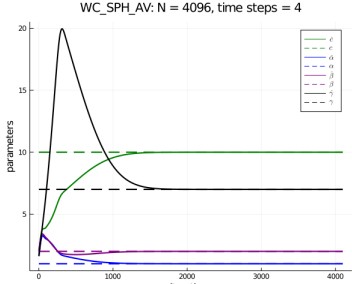 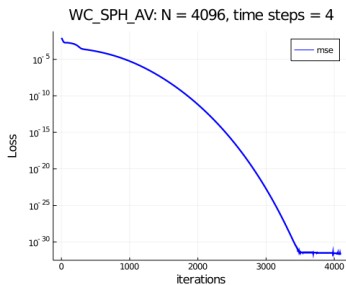

Figure 2: Solving an inverse problem for the fully physics informed model on 3D SPH flow with deterministic external forcing on 4096 particles over 4 SPH parameters. The solid lines show the SPH model parameters (initially chosen to be uniformly distributed about (0,1)) converging to the dashed lines representing the ground truth parameters. Here the field based loss function (see subsection 4.4) is used and is converging up to the order of machine precision (as Float64 and the MSE of field difference was used). See Figure 6, and Figure 7 for similar results.

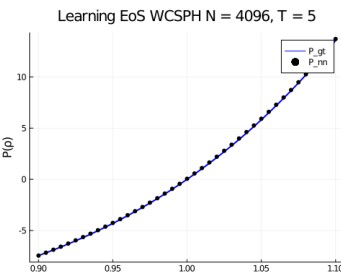 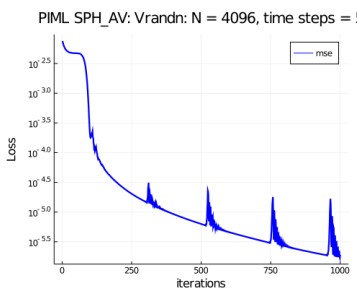

Figure 3: Using a Neural Network to approximate the equation of state (see subsection 4.1) using the field based loss function. We see that $P(\rho)$ is well approximated, where $P_{gt}$ is the ground truth EoS and $P_{nn}$ is the neural network approximation. We note that the underlying ground truth data has density variations within 1% of mean density, so the NN sufficiently approximates the EoS over a larger domain of densities as seen in training (see Figure 8 for another example).

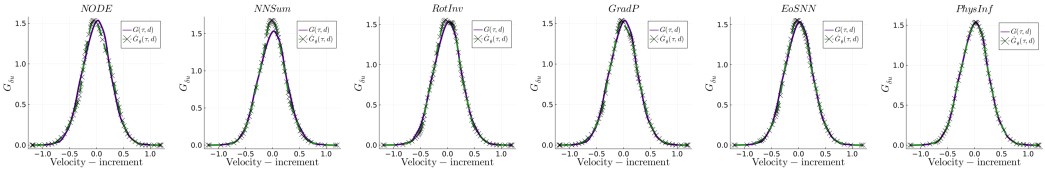

Figure 4: The velocity increment distribution (in $u$, i.e $G_{\delta u}$) learned over each model in the hierarchy subsection 4.1. The dashed crossed green line is the predicted velocity increment distribution at the specific iteration $\hat{G}_{\boldsymbol{\theta}}$, and the solid purple line is the ground truth velocity increment distribution. Each model is learned using the $L_{kl} + L_f$ loss and the FSA method. From left ($NODE$) to right (Fully informed $PhysInf$), more physical structure is added. See Figure 5 for further comparisons of these learned models

## 5.2 GENERALIZABILITY (EXTRAPOLATION CAPABILITY)

When the training is complete (i.e. when the loss function reaches its minimum) we validate extrapolation capability of the models on test data sets which are longer in duration or derived from the setting corresponding to stronger turbulence (which we control by increasing intensity of the injection term, $\boldsymbol{f}_{ext}$, while keeping the integral, i.e. energy injection scale, constant).

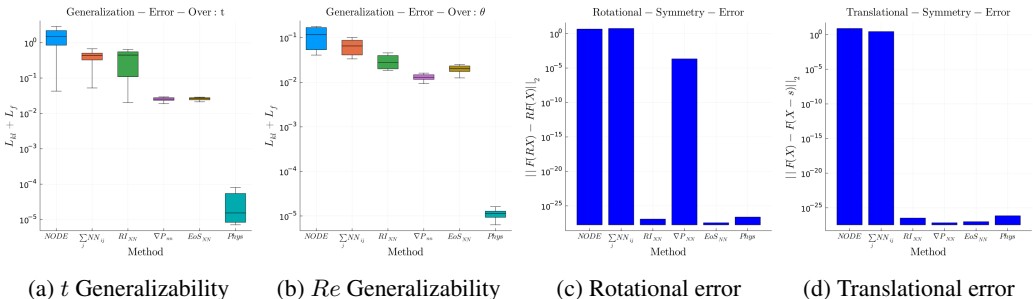

(a) $t$ Generalizability     (b) $Re$ Generalizability     (c) Rotational error     (d) Translational error

Figure 5: (a) Box plot using 15 different generalization errors over physically relevant time scales ranging from the shortest, $t_\lambda$ (which is the time it takes for a pair of initially neighboring particles to double their relative separation), up to the longest, $t_{eddy}$ (which is the time it takes for the largest eddy of the flow to make a complete revolution). (b) Box plot over 5 different generalization errors over different Reynolds numbers, ranging in values up to 3 times larger than training data where the time scale is fixed at $t_\lambda$. (c) Error in rotational and (d) translational symmetries in the learned models (i.e measuring conservation of angular and linear momentum of learned models respectively).

We observe in Figure 5, showing results for trained models using $L_f + L_{kl}$, that the richer (physical) structure added to the model is, the lower the generalization error becomes when we extrapolate to flows with larger $Re$ numbers (and over different time scales). Further experiments were conducted on two dimensional flows (see Figure 10, shown in the Appendix; here, we observe similar trends in generalization). Furthermore, Figure 5 shows that as more physical structure is added to the model, the error in rotational and translational invariance decreases (and therefore conservation of angular and linear momentum is preserved, even though it was not enforced directly). Finally, we see additional qualitative generalizability results in Figure 9 (shown in the Appendix) which allows us to conclude that data (samples of particle trajectories) of much shorter (temporal) duration are required to train models which are more physics informed (where a short time duration, $t_\lambda$, is used in training, and we see the converged models extrapolating to longer time scales not seen in training). Figure 9 also shows, qualitatively, that the large scales structures of the flows are well preserved as we extrapolate for longer times than used in training.

## 6 CONCLUSIONS

Combining modern tools in CFD, machine learning, deep learning, automatic differentiation, and classical sensitivity analysis, we have developed a learn-able hierarchy of parameterized "physics-explainable" Lagrangian fluid simulators and showed that adding physical structure improves inter-pretability, generalizability (over different time scales and $Re$ numbers), and requires less training data compared to less informed models (such as Neural ODE). Additionally, we see that as more of the physical structure is hard codded into the SPH based models, the better is its ability to conserve linear and angular momentum (translational and rotational symmetries respectively) which is known to be the case with the underlying ground truth flow.

Furthermore, these parameterized SPH simulators can be used to solve inverse problems (in the physically interpretable parameter space as well as in the NN parameter space), fit underlying flow data using a combination of field based and statistical based loss functions, and be used to learn unknown functions embedded within SPH based models. We showed that each model is capable of learning the parameters so that the underlying velocity increment distribution (a single particle Lagrangian statistics) is approximated.

In the future, we plan to improve beyond what is accomplished in this manuscript, that is to solve efficiently inverse Lagrangian problems with SPH. Specifically, we aim to build Reduced-Order Lagrangian Models of Turbulence based on the SPH approach which provide the best fit to turbulence data coming from more realistic simulations and/or experiments. It is of a special interest to push the PIML based Lagrangian SPH approach towards discovering reduced turbulence models which will extrapolate into difficult regimes, e.g. corresponding to regimes of high $Re$ number and stronger degree of compressibility, where the data are limited.

## 7 REPRODUCIBILITY STATEMENT

Supplementary material is provided that includes the Julia source code of our parameterized SPH simulators, the gradient based learning algorithm, and sensitivities for each model in our hierarchy. The general steps to take in order to reproduce these results are as follows: generate SPH ground truth flows using the parameters described in section 3 (or use the data-sets provided), then use this data for learning the hierarchy of models found in the main.jl file (under the 3d-phys-semi-inf directory) which requires a selection of the model, loss function, sensitivity method, number of iterations, and height (where height is found through tuning to have a value of $5$, other than for the EoS model which performs better with a height greater than $8$. Note, with these heights, most of the models containing NNs have a total number of parameters on the order $p \sim \mathcal{O}(100)$; in this case FSA is usually fastest (ASA is slightly faster for the NODE model, as it involves more parameters than other models due to non-pairwise interaction of particles). However, for consistency, we have selected the FSA method when comparing all models with this relatively small number of parameters. The saved models are provided in the output data files which are used in post-processing. Furthermore, in the final version of this manuscript a link to the github repository will be included.

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

## A  APPENDIX: SPH (IN MORE DETAILS)

### A.1  BASIC FORMULATION

Essentially, SPH is a discrete approximation to a continuous flow field by using a series of discrete particles. Lets first start with the identity

$$A(\mathbf{r}) = \int_V A(\mathbf{r}')\delta(\mathbf{r} - \mathbf{r}')d\mathbf{r}', \tag{10}$$

where $A$ is any scalar or tensor field. Using the smoothing kernel $W$ (for interpolation onto smooth "blobs" of fluid) and after a Taylor expansion it can be shown (according to symmetry of smoothing kernel Cossins (2010)) that

$$A(\mathbf{r}) = \int_V A(\mathbf{r}')W(|\mathbf{r} - \mathbf{r}'|, h)d\mathbf{r}' + \mathcal{O}(h^2). \tag{11}$$

where $W$ is constrained to behave similar to the delta function,

$$\int_V W(\mathbf{r}, h)d\mathbf{r} = 1, \quad \lim_{h \to 0} W(\mathbf{r}, h) = \delta(\mathbf{r}).$$

The choice of smoothing kernels is important, and effects the consistency and accuracy of results Monaghan (2005), where bell-shaped, symmetric, monotonic kernels are the most popular Fulk & Quinn (1996), however there is still disagreement on the best smoothing kernels to use. Commonly used are the B-spline smoothing kernels with a finite support (approximating a Gaussian kernel). The cubic smoothing kernel is used in this work has the following form:

$$w(q) = \sigma \begin{cases} \frac{1}{4}(2-q)^3 - (1-q)^3 & 0 \le q < 1 \\ \frac{1}{4}(2-q)^3 & 1 \le q \le 2 \\ 0 & 2 \le q \end{cases} \tag{12}$$

where, $W(|\boldsymbol{r} - \boldsymbol{r}_j|, h) = h^{-d}w(q)$ with $q = |\boldsymbol{r} - \boldsymbol{r}_j|/h$, and $\sigma = \sigma(d) = [1/\pi$ if $d = 3,\ 10/7\pi$ if $d = 2]$ is a normalizing constant to satisfy the integral constraint on $W$ (see Cossins (2010); Fulk & Quinn (1996); Monaghan (2012) for more details). The finite support allows one to use neighborhood list algorithms discussed below to utilize computational advantages (only requires a local cloud of interacting particles instead of all the particles in the computational domain that would be required with a Gaussian kernel).

SPH can be formulated through approximating integral interpolants of any scalar or tensor field $A$ by a series of discrete particles

$$\langle A(\mathbf{r}) \rangle = \int_V A(\mathbf{r}')W(|\mathbf{r} - \mathbf{r}'|, h)\mathbf{dr}' \approx \sum_i m_i \frac{A(\mathbf{r}_i)}{\rho(\mathbf{r}_i)}W(|\mathbf{r} - \mathbf{r}_i|, h), \tag{13}$$

(i.e a convolution of $A$ with $W$) where $\mathbf{dr}'$ denotes a volume element and $W(\boldsymbol{q}, h)$ is the smoothing kernel. Each particle represents a continuous "blob" of fluid and carries the fluid quantities in the Lagrangian frame (such as pressure $P_i$, density $\rho_i$, velocity $\boldsymbol{v}_i$, etc.)

The convenience of this method becomes apparent when the differential operators are approximated, i.e.

$$\nabla_{\boldsymbol{r}} \langle A(\mathbf{r}) \rangle = \frac{\partial}{\partial \mathbf{r}} \int_V A(\mathbf{r}')W(||\mathbf{r} - \mathbf{r}'||_2, h)dr' \approx \sum_i m_i \frac{A(\mathbf{r}_i)}{\rho(\mathbf{r}_i)}\nabla_{\boldsymbol{r}}W(||\mathbf{r} - \mathbf{r}_i||_2, h)), \tag{14}$$

where we see that in this direct approach to approximate the gradient operator we only need to know the gradient of the smoothing kernel (which is usually fixed beforehand). Multiple methods have been proposed and different methods are best suited for different problems. Similar approximations hold for taking the divergence or curl of a vector field Gingold & Monaghan (1977). The most common approximation of the gradient operator is derived from the following identity,

$$\nabla_{\boldsymbol{r}}A(\boldsymbol{r}_i) = \rho\left(\frac{A(\boldsymbol{r}_i)}{\rho^2}\nabla_{\boldsymbol{r}}\rho - \nabla_{\boldsymbol{r}}\left(\frac{A(\boldsymbol{r}_i)}{\rho}\right)\right), \tag{15}$$

and is approximated with particles as

$$\nabla_{\boldsymbol{r}}A_i \approx \rho_i \sum_j^N m_j \left(\frac{A_j}{\rho_j^2} - \frac{A_i}{\rho_i^2}\right)\nabla_{\boldsymbol{r}_i}W_{ij}. \tag{16}$$

### A.2 Approximation of Flow Equations

The above integral interpolant approximations using series of particles can be used to discretize the equations of motion (as seen in Eq. (1) with more details found in Gingold & Monaghan (1977); Monaghan (2012). Each particle carries a mass $m_i$ and velocity $\boldsymbol{v}_i$, and other properties (such as pressure, density etc.). We can use Eq. (13) to estimate the density everywhere by

$$\rho(\boldsymbol{r}_i) = \sum_j m_j W(|\boldsymbol{r}_i - \boldsymbol{r}_j|, h), \tag{17}$$

where although the summation is over all particles, because the smoothing kernel has compact support the summation only needs to occur over the smoothing radius (here $2h$ as seen in Eq. (12). Another popular way to approximate the density is through using the continuity equation and approximating the divergence of the velocity field in different ways Monaghan (1992). In what follows we use the notation $A_i = A(\boldsymbol{r}_i)$. Using the gradient approximation defined above (Eq. (16), the pressure gradient could be estimated by using

$$\rho_i \nabla_{\boldsymbol{r}}P_i = \sum_j m_j(P_j - P_i)\nabla_{\boldsymbol{r}_i}W_{ij},$$

where $W_{ij} = W(|\boldsymbol{r}_i - \boldsymbol{r}_j|, h)$. However, in this form the momentum equation $d_t\boldsymbol{v} = -\frac{1}{\rho}\nabla_{\boldsymbol{r}}P$ does not conserve linear and angular momentum Monaghan (1992). To improve this, a symmetrization is

often done to the pressure gradient term by rewriting $\frac{\nabla_r P}{\rho} = \partial_r \left( \frac{P}{\rho} \right) + \frac{P}{\rho^2} \nabla_r \rho$. This results in a momentum equation for particle $i$ discretized as

$$\frac{d\boldsymbol{v}_i}{dt} = -\sum_j m_i \left( \frac{P_j}{\rho_j^2} + \frac{P_i}{\rho_i^2} \right) \nabla_{\boldsymbol{r}_i} W_{ij},$$

which produces a symmetric central force between pairs of particles and as a result linear and angular momentum are conserved Monaghan (1992). Next, including an artificial viscosity term and external forcing, the full set of ODEs approximating PDEs governing fluid motion is

$$\frac{d\mathbf{r}_i}{dt} = \mathbf{v}_i \quad \forall i \in \{1, 2, ...N\} \tag{18}$$

$$\frac{d\mathbf{v}_i}{dt} = -\sum_{j \neq i}^{N} m_j \left( \frac{P_j}{\rho_j^2} + \frac{P_i}{\rho_i^2} + \Pi_{ij} \right) \nabla_{\boldsymbol{r}_i} W_{ij} + \boldsymbol{f}_{ext} \quad \forall i \in \{1, 2, ...N\}. \tag{19}$$

In this work, we start by using the weakly compressible formulation by assuming a barotropic fluid, where equation of state (EoS) is given by

$$P(\rho) = \frac{c_s^2 \rho_0}{\gamma} \left[ \left( \frac{\rho}{\rho_0} \right)^\gamma - 1 \right], \tag{20}$$

as in Monaghan (2012), where $\rho_0$ is the initial reference density, and $\gamma = 7$ is used. In future work, we plan on including the energy equation to extend these methods for highly compressible applications.

There are many different forms of artificial viscosity that have been proposed Monaghan (1997). In this work, we use the popular formulation of $\Pi_{ij}$ that approximates the contribution from the bulk and shear viscosity along with an approximation of Nueman-Richtmyer viscosity for handling shocks Monaghan (2012); Morris et al. (1997):

$$\Pi_{ij} = \begin{cases} \dfrac{-\alpha c \mu_{ij} + \beta \mu_{ij}^2}{\rho_{ij}}, & \mathbf{v}_{ij} \cdot \mathbf{r}_{ij} < 0 \\ 0, & \text{otherwise} \end{cases}, \tag{21}$$

where $c$ represents the speed of sound and

$$\mu_{ij} = \frac{h\mathbf{v}_{ij} \cdot \mathbf{r}_{ij}}{|\mathbf{r}_{ij}|^2 + \epsilon h^2}, \quad \rho_{ij} = 0.5(\rho_i + \rho_j),$$

This artificial viscosity term was constructed in the standard way following Monaghan (1997); Monaghan & Gingold (1983): The linear term involving the speed of sound was based on the viscosity of an ideal gas. This term scales linearly with the velocity divergence, is negative to enforce $\Pi_{ij} > 0$, and should be present only for convergent flows ($\boldsymbol{v}_{ij} \cdot \boldsymbol{r}_{ij} < 0$). The quadratic term including $(\boldsymbol{v}_{ij} \cdot \boldsymbol{r}_{ij})^2$ is used to prevent penetration in high Mach number collisions by producing an artificial pressure roughly proportional to $\rho|\boldsymbol{v}|^2$ and approximates the von Neumann-Richtmyer viscosity (and should also only be present for convergent flows). There are several advantages to this formulation of $\Pi_{ij}$; mainly it is Galilean and rotationally invariant, thus conserves total linear and angular momentum. A more detailed derivation is found in Cossins (2010) (along with other formulations of artificial viscosities).

In practice the summation Eq. (1) over all particles is carried out through a neighborhood list algorithm (such as the cell linked list algorithm with a computational cost that scales as $\mathcal{O}(N)$ Domínguez et al. (2011)). We also note that Eq. (1) can also be derived from Euler-Lagrange equations after defining a Lagrangian, see Cossins (2010) (for respective analysis of the inviscid case, when the artificial viscosity term is neglected), then an artificial viscosity $\Pi_{ij}$ term can be incorporated from using SPH discretizations, see Monaghan (2005) for details.

## A.3 External Forcing

In order to approach a stationary homogeneous and isotropic turbulent flow, we use a combination of external forcing, both stochastic and deterministic. We first use a stochastic external forcing acting on large scales which forces 10 modes that are close to the origin in Fourier space. Formally,

$$\boldsymbol{f}_{ext} = \begin{pmatrix} f^x_{ext} \\ f^y_{ext} \end{pmatrix}, \quad f^x_{ext}[i] = -\sum_{k=1}^{10} \cos(f^x_k x_i + f^y_k y_i + 2\pi\zeta(t,k))\tilde{f}_k f^y_k,$$

$$f^y_{ext}[i] = \sum_{k=1}^{10} \cos(f^x_k x_i + f^y_k y_i + 2\pi\zeta(t,k))\tilde{f}_k f^x_k, \quad \tilde{f}_k = 0.25\left(\frac{2.5 - \sqrt{(f^x_k)^2 + (f^y_k)^2}}{1.5}\right)^2,$$

where $(f^x_k, f^y_k)$ are the frequency components that lie near to the origin (in frequency space), and $\zeta(t,k)$ is a stochastic white noise term.

Once the flow achieves stationarity, a deterministic forcing is used (for simplifying the learning algorithms described in the next section), which is commonly used in CFD literature, e.g. (Petersen & Livescu, 2010) for analyzing stationary homogeneous and isotropic turbulence. Then,

$$\boldsymbol{f}_{ext}[i] = \frac{\theta}{\boldsymbol{ke}}(\boldsymbol{v}_i - \bar{\boldsymbol{v}}), \quad \boldsymbol{ke} = \frac{0.5}{N}\sum_{k=1}^{N}(u_k^2 + v_k^2)$$

is the kinetic energy computed at each time step, $\theta$ represents the rate of energy injected into the flow and $\bar{\boldsymbol{v}}$ is the mean velocity for each component. $\theta$ is a fixed quantity which is set before hand, and the flow is integrated until stationarity is reached.

## A.4 Numerical Algorithm for Forward Solving SPH

We use a symplectic integrator for generating the "synthetic" ground truth data, and for making prediction steps required in our gradient based optimization described in Eq. (4. We apply the Verlet integration scheme (leap-frog) to solve the Initial Value Problem. Using the notation,

$$\boldsymbol{X} = \{(\boldsymbol{r_i}, \boldsymbol{v_i})|\forall i \in \{1, ..., N\}\}, \quad \boldsymbol{\rho} = \{\boldsymbol{\rho}_i|\forall i \in \{1, ..., N\}\}, \quad \frac{d\boldsymbol{r}_i}{dt} = \boldsymbol{v}_i, \quad \frac{d\boldsymbol{v}_i}{dt} = \boldsymbol{F}_i(\boldsymbol{\rho}, \boldsymbol{X}),$$

we proceed according to the following algorithm

1: Compute $\boldsymbol{\rho}^k$ using Eq. (13),
2: Compute $\boldsymbol{F}_i^k(\boldsymbol{\rho}^k, \boldsymbol{X}^k)$ using Eq. (1),
3: $\boldsymbol{v}_i^{k+\frac{1}{2}} = \boldsymbol{v}_i^k + \frac{\Delta t}{2}\boldsymbol{F}_i^k$,
4: $\boldsymbol{r}_i^{k+1} = \boldsymbol{r}_i^k + \Delta t \boldsymbol{v}_i^{k+\frac{1}{2}}$,
5: Compute $\boldsymbol{\rho}^{k+1}$ using Eq. (13),
6: Compute $\boldsymbol{F}_i^{k+\frac{1}{2}}(\boldsymbol{\rho}^{k+1}, \boldsymbol{X}^{k+\frac{1}{2}})$ using Eq. (1),
7: $\boldsymbol{v}_i^{k+1} = \boldsymbol{v}_i^{k+\frac{1}{2}} + \frac{\Delta t}{2}\boldsymbol{F}_i^{k+\frac{1}{2}}$,

repeated for each time step, $k \in \{0, \Delta t, ..., T\}$, where the time step, $\Delta t$, is chosen according to the Courant-Friedrichs-Lewy (CFL) condition, $\Delta t \leq 0.4h/c$. This algorithm has the following physical interpretation: it prevents spatial information transfer through the code at a rate greater than the local speed of sound (small in the almost incompressible case considered in this manuscript).

## B Appendix: Methods

### B.0.1 Forward Sensitivity Analysis

In general, the loss functions in this work can be defined as

$$L(\boldsymbol{X}, \boldsymbol{\theta}) = \int_0^{t_f} \Psi(\boldsymbol{X}, \boldsymbol{\theta}, t)dt. \tag{22}$$

The forward SA (FSA) approach simultaneously integrates the state variables along with their sensitivities (with respect to parameters) forward in time to compute the gradient of $L$;

$$d_{\boldsymbol{\theta}} L(\boldsymbol{X}, \boldsymbol{\theta}) = \int_0^{t_f} \partial_{\boldsymbol{X}} \Psi(\boldsymbol{X}, \boldsymbol{\theta}, t) d_{\boldsymbol{\theta}} \boldsymbol{X}(\boldsymbol{\theta}, t) + \partial_{\boldsymbol{\theta}} \Psi(\boldsymbol{X}, \boldsymbol{\theta}, t) dt. \tag{23}$$

Where, through using the chain rule, we see that the sensitivities of the state variables with respect to the model parameters ($d_{\boldsymbol{\theta}} \boldsymbol{X}$) are required to compute the gradient of the loss. Assuming that the initial conditions of the state variables do not depend on the parameters, then $\partial \boldsymbol{X}(0)/\partial \theta^{\alpha} = 0$. Now, define the sensitivities as $\boldsymbol{S}_i^{\alpha} := d\boldsymbol{X}_i/d\theta^{\alpha}$. Then, from Eq. (8) we derive

$$\frac{d\boldsymbol{S}_i^{\alpha}}{dt} = \frac{d\boldsymbol{\mathcal{F}}_i(\boldsymbol{X}(t), \boldsymbol{\theta})}{d\theta^{\alpha}}, \tag{24}$$

then resulting, after applying the chain rule, in

$$\frac{d\boldsymbol{S}_i^{\alpha}}{dt} = \frac{\partial \boldsymbol{\mathcal{F}}_i(\boldsymbol{X}(t), \boldsymbol{\theta})}{\partial \boldsymbol{X}_i} \boldsymbol{S}_i^{\alpha} + \frac{\partial \boldsymbol{\mathcal{F}}_i(\boldsymbol{X}(t), \boldsymbol{\theta})}{\partial \theta^{\alpha}}. \tag{25}$$

Since the initial condition $\boldsymbol{X}(0)$ does not depend on $\boldsymbol{\theta}$, then $\boldsymbol{S}_i^{\alpha}(0) = 0$. Now, computing the gradient of the loss function reduces to solving a forward in time Initial Value Problem (IVP) by integrating simultaneously the state variables $\boldsymbol{X}_i$ defined in the main text, and sensitivities $\boldsymbol{S}_i^{\alpha}$, defined in Eq. (25). However, this means that the computational cost of the FSA method scales with $\mathcal{O}(Nk)$, where $k$ is the number of parameters, which can be large when using NN's. With $\boldsymbol{S}_i^{\alpha}$ known, the parameters can then be learned (estimated) through an iterative first order gradient based approach (such as by updating $\boldsymbol{\theta}$ using adaptive moment estimation (Adam) (Kingma & Ba, 2017)) where the gradient is computed in Eq. (23).

In order to integrate Eq. (25) the gradient of $\boldsymbol{\mathcal{F}}_i$ with respect to the parameters, both $\partial \boldsymbol{\mathcal{F}}_i(\boldsymbol{X}(\tau), \boldsymbol{\theta})/\partial \theta^{\alpha}$ and the Jacobian matrix, $\{\partial \boldsymbol{\mathcal{F}}_i(\boldsymbol{X}(\tau), \boldsymbol{\theta})/\partial x_j | \forall i, j\}$ need to be computed. In this work, this is done with a mixed mode approach. $\partial \boldsymbol{\mathcal{F}}_i(\boldsymbol{X}(\tau), \boldsymbol{\theta})/\partial \theta^{\alpha}$, with $\boldsymbol{\mathcal{F}}_i(\boldsymbol{\theta}) : \mathbb{R}^k \to \mathbb{R}^{2d}$, is computed with AD (the choice of forward or reverse mode is determined by the dimension of the input and output space), where $k$ is the number of parameters and $d$ is the dimension. For example, if $k \gg 2d$ (as is the case when $NN$s are used), reverse mode AD is more efficient than forward mode (Ma et al., 2021). The Jacobian matrix is computed and obtained through mixing symbolic differentiation packages (or analytically deriving by hand), as well as mixing AD. For example, when there are NNs used for the parameterization of the right hand side, then according to expression for the Jacobian from the main text, AD derivatives will need to be computed on different functions each with potentially different dimensions of input and output space. With $\boldsymbol{\mathcal{F}}_i(\boldsymbol{X}) : \mathbb{R}^{2d} \to \mathbb{R}^{2d}$, forward mode AD is more efficient. As seen in the subsection 4.1, some of the models we train require neighborhood search list algorithms, which in this case AD is only applied to derivative within the Jacobian, especially whenever summation over particles is done. Thus, a mixed mode approach is taken by combining both forward and reverse mode (depending on dimensions) to improve the efficiency over just applying one or the other. The AD packages used in this work were both ForwardDiff.jl (Revels et al., 2016) for forward mode and Zygote.jl (Innes, 2018) for reverse mode.

$$\frac{\partial \mathcal{F}_i(\boldsymbol{X}(t), \boldsymbol{\theta})}{\partial \boldsymbol{X}_i} = \begin{pmatrix} [0]_2 & I_2 \\ \dfrac{\partial \boldsymbol{F}_i(\boldsymbol{X}(t), \boldsymbol{\theta})}{\partial \boldsymbol{X}_i} \end{pmatrix} = \begin{pmatrix} [0]_2 & I_2 \\ \dfrac{\partial F_i^x}{\partial x_i^1} & \dfrac{\partial F_i^x}{\partial x_i^2} & \dfrac{\partial F_i^x}{\partial x_i^3} & \dfrac{\partial F_i^x}{\partial x_i^4} \\ \dfrac{\partial F_i^y}{\partial x_i^1} & \dfrac{\partial F_i^y}{\partial x_i^2} & \dfrac{\partial F_i^y}{\partial x_i^3} & \dfrac{\partial F_i^y}{\partial x_i^4} \end{pmatrix}. \tag{26}$$

### B.0.2 ADJOINT METHOD

This section provides an outline of the Adjoint SA (ASA) method used in this work (and can be found in (Bradley, 2019) (Donello et al., 2020)). Again, the goal is to compute the gradient of the loss function. This is a continuous time dependent formulation, where the goal is to minimize a

loss function $L(\boldsymbol{X}(\boldsymbol{\theta},t),\boldsymbol{\theta})$ which is integrated over time, $L(\boldsymbol{X},\boldsymbol{\theta}) = \int_0^{t_f} \Psi(\boldsymbol{X},\boldsymbol{\theta},t)dt$, subject to the physical structure constraints (ODE or PDE), $H(\boldsymbol{X},\dot{\boldsymbol{X}},\boldsymbol{\theta},t) = 0$, and the dependence of the initial condition, $g(\boldsymbol{X}(0),\boldsymbol{\theta}) = 0$, on parameters. Here, $H$ is the explicit ODE in the standard form obtained through the SPH discretization of the PDE flow equations Eq. (8),

$$H(\boldsymbol{X},\dot{\boldsymbol{X}},\boldsymbol{\theta},t) = \dot{\boldsymbol{X}}(t) - \boldsymbol{\mathcal{F}}(\boldsymbol{X}(t),\boldsymbol{\theta}). \tag{27}$$

A gradient based optimization algorithm requires that the gradient of the loss function,

$$d_{\boldsymbol{\theta}}L(\boldsymbol{X},\boldsymbol{\theta}) = \int_0^{t_f} \partial_{\boldsymbol{X}}\Psi(\boldsymbol{X},\boldsymbol{\theta},t)d_{\boldsymbol{\theta}}\boldsymbol{X}(\boldsymbol{\theta},t) + \partial_{\boldsymbol{\theta}}\Psi(\boldsymbol{X},\boldsymbol{\theta},t)dt,$$

be computed. The main difference in the FSA and ASA approach is that in the ASA calculating $d_{\boldsymbol{\theta}}\boldsymbol{X}$ is not required (which avoids integrating the additional $k$ ODEs as in FSA). Instead, the adjoint method develops a second ODE (size of which is independent of $k$) in the adjoint variable $\boldsymbol{\lambda}$ as a function of time (which is then integrated backwards in time).

The following provides a Lagrange multiplier approach to deriving this ODE in $\boldsymbol{\lambda}$. First define

$$\mathcal{L} = \int_0^{t_f} (\Psi(\boldsymbol{X},\boldsymbol{\theta},t) + \boldsymbol{\lambda}^T(t)H(\boldsymbol{X},\dot{\boldsymbol{X}},\boldsymbol{\theta},t))dt + \mu^T g(\boldsymbol{X}(0),\boldsymbol{\theta}) \tag{28}$$

where $\boldsymbol{\lambda}$ and $\mu$ are the Lagrange multipliers. Now, since $H$ and $g$ are zero everywhere, we may choose the values of $\boldsymbol{\lambda}$ and $\mu$ arbitrarily. Then, $\nabla_{\boldsymbol{\theta}}\mathcal{L} = \nabla_{\boldsymbol{\theta}}L$, resulting in

$$\begin{aligned} \nabla_{\boldsymbol{\theta}}\mathcal{L} &= \int_0^{t_f} \left( \partial_{\boldsymbol{X}}\Psi d_{\boldsymbol{\theta}}\boldsymbol{X} + \partial_{\boldsymbol{\theta}}\Psi + \boldsymbol{\lambda}^T(\partial_{\boldsymbol{X}}H d_{\boldsymbol{\theta}}\boldsymbol{X} + \partial_{\dot{\boldsymbol{X}}}H d_{\boldsymbol{\theta}}\dot{\boldsymbol{X}} + \partial_{\boldsymbol{\theta}}H) \right) dt \\ &+ \mu^T(\partial_{\boldsymbol{X}(0)}g d_{\boldsymbol{\theta}}\boldsymbol{X}(0) + \partial_{\boldsymbol{\theta}}g). \end{aligned}$$

Integrating the equation by parts, and elimitating, $d_{\boldsymbol{\theta}}\dot{\boldsymbol{X}}$, we arrive at

$$\begin{aligned} \nabla_{\boldsymbol{\theta}}\mathcal{L} &= \int_0^{t_f} \left[ \left( \partial_{\boldsymbol{X}}\Psi + \boldsymbol{\lambda}^T(\partial_{\boldsymbol{X}}H - d_t\partial_{\dot{\boldsymbol{X}}}H) - \dot{\boldsymbol{\lambda}}^T\partial_{\dot{\boldsymbol{X}}}H \right) d_{\boldsymbol{\theta}}\boldsymbol{X} + \partial_{\boldsymbol{\theta}}\Psi + \boldsymbol{\lambda}^T\partial_{\boldsymbol{\theta}}H \right] dt \\ &+ \boldsymbol{\lambda}^T\partial_{\dot{\boldsymbol{X}}}H d_{\boldsymbol{\theta}}\boldsymbol{X}\Big|_{t_f} + (-\boldsymbol{\lambda}^T\partial_{\dot{\boldsymbol{X}}}H + \mu^T g)\Big|_0 d_{\boldsymbol{\theta}}\boldsymbol{X}(0) + \mu^T\partial_{\boldsymbol{\theta}}g. \end{aligned}$$

Since the choice of, $\boldsymbol{\lambda}^T$, and, $\mu$, is arbitrary, we set, $\boldsymbol{\lambda}^T(T) = 0$, and, $\mu^T = (\boldsymbol{\lambda}^T\partial_{\dot{\boldsymbol{X}}}H)|_0(g|_{\boldsymbol{X}(0)})^{-1}$, in order to avoid needing to compute, $d_{\boldsymbol{\theta}}\boldsymbol{X}(T)$, and thus canceling the second to the last term in the latest (inline) expression. Now, assuming that the initial values of the state variables, $\boldsymbol{X}(0)$, do not depend on the parameters, we derive, $d_{\boldsymbol{\theta}}\boldsymbol{X}(0) = 0$, and, $g = 0$. Then, we use a loss function $\Psi$ that does not depend on $\boldsymbol{\theta}$ explicitly, so that, $\partial_{\boldsymbol{\theta}}\Psi = 0$. And finally, we can avoid computing $d_{\boldsymbol{\theta}}\boldsymbol{X}$ at all other times $t > 0$ by setting

$$\partial_{\boldsymbol{X}}\Psi + \boldsymbol{\lambda}^T(\partial_{\boldsymbol{X}}H - d_t\partial_{\dot{\boldsymbol{X}}}H) - \dot{\boldsymbol{\lambda}}^T\partial_{\dot{\boldsymbol{X}}}H = 0. \tag{29}$$

The resulting equation for the time derivative of $\boldsymbol{\lambda}$ can be re-stated as the following adjoint ODE

$$\dot{\boldsymbol{\lambda}}^T = \partial_{\boldsymbol{X}}\Psi - \boldsymbol{\lambda}^T\frac{\partial\boldsymbol{\mathcal{F}}}{\partial\boldsymbol{X}}, \qquad \boldsymbol{\lambda}(t_f) = 0, \tag{30}$$

where we also used that according to Eq. (27), $\partial_{\boldsymbol{X}}H = -\partial_{\boldsymbol{X}}\boldsymbol{\mathcal{F}}$ and $\partial_{\dot{\boldsymbol{X}}}H = I_{2d}$.

Combining all of the above, we see that the simplified equation for the gradient of the loss function is

$$\nabla_{\boldsymbol{\theta}}L = \nabla_{\boldsymbol{\theta}}\mathcal{L} = \int_0^{t_f} \boldsymbol{\lambda}^T\partial_{\boldsymbol{\theta}}H dt.$$

Which, according to Eq. (27), becomes

$$\nabla_{\boldsymbol{\theta}}L = -\int_0^{t_f} \boldsymbol{\lambda}^T\frac{\partial\boldsymbol{\mathcal{F}}}{\partial\boldsymbol{\theta}}dt. \tag{31}$$

Therefore, in order to compute the gradient, the IVP expression (30) needs to be integrated backwards in time for $\boldsymbol{\lambda}(t)$, starting from $\boldsymbol{\lambda}(t_f) = 0$. Similar to the FSA formulation found above, both $\partial_{\boldsymbol{X}}\mathcal{F}$ and $\partial_{\boldsymbol{\theta}}\mathcal{F}$, are computed with a mixture of forward and reverse mode AD tools, depending on the dimension of the input and output dimension of the functions to be differentiated.

Although the adjoint method has a computational cost that is independent of the number of parameters, it requires more memory as the adjoint Eq. (30) must be solved backward in time (and requires the forward solution to be stored). Notice that this workflow is not adequate for problems where real-time sensitivities are needed, as discussed in (Donello et al., 2020). This usually means that for $k \sim \mathcal{O}(100)$ FSA is more efficient but when $k \gg 100$, the situation reverses and ASA becomes more efficient. See (Ma et al., 2021) for more details on the difference in efficiencies of mixing FSA (or ASA) with forward and reverse mode AD.

### B.1 LOSS FUNCTIONS

In the previous subsections, we introduced the FSA and ASA methods for computing the gradient of the loss function, which is used in our gradient based optimization learning algorithms. In this section we construct three different loss functions; trajectory based, field based, and statistical based. The trajectory based loss function tries to minimize the difference in trajectories of the predicted particles from that of the ground truth trajectories. The field based loss function tries to minimize the difference in the velocity fields of the predicted and ground truth data (more fitting for the goal of SPH: to approximate the field quantities). The statistical based loss function tries to minimize the difference of underlying probability distributions obtained from the predicted and ground truth data. We explored all three loss functions and combinations of each in this work. However, since our overall goal involves learning SPH models for turbulence applications, it is the underlying statistical description we want our models to learn and generalize to. This is discussed further in section 5 where we compare the hierarchy of models that is constructed in the next subsection 4.1

#### B.1.1 TRAJECTORY BASED LOSS

A simple loss function to consider is the Mean Squared Error ($MSE$) of the difference in particles and velocity:

$$MSE(\theta) = \frac{1}{N_f}\|\boldsymbol{X} - \hat{\boldsymbol{X}}\|^2,$$

where $\boldsymbol{X}$ and $\hat{\boldsymbol{X}}$ are the ground truth and predicted particles states (both position and velocity) over the $N$ particles respectively.

#### B.1.2 FIELD BASED LOSS

The field based loss function tries to minimize the difference of the large scale structures found in the fields of which the particles are used to approximate. This is obtained by interpolating the velocities of the particles onto a mesh as seen in the following.

$$L_f(\theta) = \frac{1}{N_f}\|\boldsymbol{V}^f - \hat{\boldsymbol{V}}^f\|^2,$$

where

$$\boldsymbol{V}_i^f = \sum_{j=1}^{N_f} \frac{m_j}{\rho_j}\boldsymbol{v}_j W_{ij}(\|\boldsymbol{r}_i^f - \boldsymbol{r}_j\|, h),$$

uses the same SPH smoothing approximation to interpolate the particle velocity onto a predefined mesh $\boldsymbol{r}^f$ (with $N_f$ grid points), and $\hat{\boldsymbol{V}}^f$ is the velocity field prediction extracted from SPH samples (multi-particle, temporal snapshots). The motivation for using a field based loss function comes from the desire to reconstruct the underlying field variables (which is what SPH is approximating for).

### B.1.3 STATISTICAL BASED LOSS

In order to approach turbulent flows, one can use well established statistical tools/objects. In this direction consider the integrated Kullback–Leibler divergence (KL) as a loss function (that will later be used in comparing a hierarchy of models).

$$L(\theta) = \int_{t=0}^{t_f} \int_{-\infty}^{\infty} G_{gt}(t, \boldsymbol{z}_{gt}(t), x) \log \left( \frac{G_{gt}(t, \boldsymbol{z}_{gt}(t), x)}{G_{pred}(t, \boldsymbol{z}_{pr}(t), x)} \right) dx dt, \tag{32}$$

where $\boldsymbol{z}_{gt}(t)$, and $\boldsymbol{z}_{pr}(t)$ represent a single particle statistical object over all particles (such as the single particle dispersion statistics or velocity increment) over time of the ground truth and predicted data respectively. Here $G(t, \boldsymbol{z}(t), x)$ is the continuous probability distribution (in $x$) constructed from data $\boldsymbol{z}(t)$ using Kernel Density Estimation (KDE). KDE is a classical unsupervised statistical learning method in order to obtain smooth and differentiable distributions from data (Chen, 2017), with

$$G(\tau, \boldsymbol{z}, x) = \frac{1}{Nh} \sum_{i=1}^{N} K\left((z_i - x)/h\right), \tag{33}$$

where $K$ is the smoothing kernel (chosen to be the normalized Gaussian in this work). Note the similarity with the SPH smoothing process, which makes it a natural choice in this setting.

The integration in time becomes a summation over discrete time steps once numerical integration over time is done (see subsection A.4) in simulating the particles and performing SA (see section 4). To simplify the resulting expression, let us fix the time $\tau$, then integration over time can be carried out last (through applying Leibniz integration rule). One derives

$$L(\theta, \tau) = \int_{-\infty}^{\infty} G_{gt}(\tau, \boldsymbol{z}_{gt}, x) \log \left( \frac{G_{gt}(\tau, \boldsymbol{z}_{gt}, x)}{G_{pred}(\tau, \boldsymbol{z}_{pr}, x)} \right) dx.$$

Combining all the preceding formulas we derive

$$L(\theta, \tau) = \int_{-\infty}^{\infty} \left[ \frac{1}{N_1 h_1} \sum_{i=1}^{N_1} K\left(\frac{z_{gt}^i - x}{h_1}\right) \right] \log \left( \frac{\frac{1}{N_1 h_1} \sum_{i=1}^{N_1} K\left(\frac{z_{gt}^i - x}{h_1}\right)}{\frac{1}{N_2 h_2} \sum_{i=1}^{N_2} K\left(\frac{z_{pr}^i - x}{h_2}\right)} \right) dx.$$

For notational convenience, let us define the integrand as $kl$

$$kl(\tau, \boldsymbol{z}_{gt}, \boldsymbol{z}_{pr}, x) = \left[ \frac{1}{N_1 h_1} \sum_{i=1}^{N_1} K\left(\frac{z_{gt}^i - x}{h_1}\right) \right] \log \left( \frac{\frac{1}{N_1 h_1} \sum_{i=1}^{N_1} K\left(\frac{z_{gt}^i - x}{h_1}\right)}{\frac{1}{N_2 h_2} \sum_{i=1}^{N_2} K\left(\frac{z_{pr}^i - x}{h_2}\right)} \right),$$

and

$$L(\theta) = \int_{-\infty}^{\infty} kl(\tau, \boldsymbol{z}_{gt}, \boldsymbol{z}_{pr}, x) dx,$$

so that

$$kl(\tau, \boldsymbol{z}_{gt}, \boldsymbol{z}_{pr}, x) : \mathbb{R} \times \mathbb{R}^{N_1} \times \mathbb{R}^{N_2} \times \mathbb{R} \to \mathbb{R}$$

.

In order to perform gradient descent, this loss function will need to be differentiated with respect to each parameter, and through the chain rule we will need the derivative of the state space variables with respect to the parameters (found through FSA or ASA mixed with AD). Some of the following derivatives are done through automatic differentiation, however, in order to not require taking derivatives through the non-differentiable neighborhood list algorithm, some of the derivative need to be done analytically (or at least symbolically).

$$\frac{\partial L}{\partial \theta^{\alpha}} = \int_{-\infty}^{\infty} \frac{\partial kl(\tau, \boldsymbol{z}_{gt}, \boldsymbol{z}_{pr}, x)}{\partial \boldsymbol{z}_{gt}} \cdot \boldsymbol{0} + \frac{\partial kl(\tau, \boldsymbol{z}_{gt}, \boldsymbol{z}_{pr}, x)}{\partial \boldsymbol{z}_{pr}} \cdot \frac{\partial \boldsymbol{z}_{pr}}{\partial \theta^{\alpha}} dx,$$

where

$$\frac{\partial kl}{\partial z_{pr_i}} = \left[ \frac{1}{N_1 h_1} \sum_{i=1}^{N_1} K\left( \frac{z_{gt}^i - x}{h_1} \right) \right] \frac{\partial}{\partial z_{pr}^i} \log \left( \frac{\frac{1}{N_1 h_1} \sum_{i=1}^{N_1} K\left( \frac{z_{gt}^i - x}{h_1} \right)}{\frac{1}{N_2 h_2} \sum_{i=1}^{N_2} K\left( \frac{z_{pr}^i - x}{h_2} \right)} \right)$$

$$= -\frac{1}{h_2} \left[ \frac{1}{N_1 h_1} \sum_{i=1}^{N_1} K\left( \frac{z_{gt}^i - x}{h_1} \right) \right] \left( \frac{\frac{1}{N_2 h_2} K'\left( \frac{z_{pr}^i - x}{h_2} \right)}{\frac{1}{N_2 h_2} \sum_{i=1}^{N_2} K\left( \frac{z_{pr}^i - x}{h_2} \right)} \right)$$

$$= -\frac{1}{h_2} G_u(\boldsymbol{z}_{gt}, x) \left( \frac{\frac{1}{N_2 h_2} K'\left( \frac{z_{pr_i} - x}{h_2} \right)}{G_u^{pred}(z_{pr}, x)} \right)$$

Using the velocity increment in both $u$ and $v$ this can be simplified:

$$\frac{\partial L}{\partial \theta^\alpha} = \int_{-\infty}^{\infty} \sum_{i=1}^{N_2} \left( \frac{\partial kl_u}{\partial z_{pr_i}^u} \frac{\partial u_{pr}^i}{\partial \theta^\alpha} + \frac{\partial kl_v}{\partial z_{pr_i}^v} \frac{\partial v_{pr}^i}{\partial \theta^\alpha} \right) dx$$

where $z_{pr_i}^u = u_{pr}^i(\tau) - u_i(0)$. Simplifying,

$$\frac{\partial L}{\partial \theta^\alpha} = \sum_{i=1}^{N_2} \left[ \frac{\partial u_{pr}^i}{\partial \theta^\alpha} \int_{-\infty}^{\infty} \frac{\partial kl_u}{\partial z_{pr_i}^u} dx + \frac{\partial v_{pr}^i}{\partial \theta^\alpha} \int_{-\infty}^{\infty} \frac{\partial kl_v}{\partial z_{pr_i}^v} dx \right],$$

where $\frac{\partial u_{pr}^i}{\partial \theta^\alpha}$, and $\frac{\partial v_{pr}^i}{\partial \theta^\alpha}$ are computed using our mixed mode approach (in section 4), $\frac{\partial kl_v}{\partial z_{pr_i}^v}$ and $\frac{\partial kl_u}{\partial z_{pr_i}^u}$ are computed with forward mode AD, and the integrals over $x$ are computed using numerical integration.

## B.2 Learning Algorithm

In what follows, we combine all of the computational tools and techniques introduced so far to outline the mixed mode learning algorithm.

---

**Algorithm 1:** Mixed Mode Learning Algorithm

---

Given Ground Truth: $\{\boldsymbol{X}(t_0), \boldsymbol{X}(t_1), ..., \boldsymbol{X}(t_f)\}$ SPH data ;
Select model: $d_t \boldsymbol{X}(t) = \boldsymbol{\mathcal{F}}(\boldsymbol{X}(t), \boldsymbol{\theta})$ from hierarchy
Select SA method (Forward or Adjoint)
Choose Loss methods (Trajectory, Field, or Probabilistic)
Choose optimizer:    e.g $RMSprop$ or $ADAM$
**for** $k \in \{1, .., n\}$ **do**

    Prediction step: Verlet integration of model $\hat{\boldsymbol{X}} = Verlet(\boldsymbol{\mathcal{F}}, \boldsymbol{\theta}_k, \boldsymbol{X}(t_0), t_0, t_f)$

    Simultaneously computes $\dfrac{\partial \boldsymbol{\mathcal{F}}}{\partial \hat{\boldsymbol{X}}_i}, \dfrac{\partial \boldsymbol{\mathcal{F}}}{\partial \boldsymbol{\theta}}$ with mixed mode AD.

    **if** *Forward SA* **then**
        Simultaneously integrate system of ODEs for sensitivities $\boldsymbol{S}_i^\alpha$ ;

$$d_t \boldsymbol{S}_i = \frac{\partial \boldsymbol{\mathcal{F}}_i(\hat{\boldsymbol{X}}(t), \boldsymbol{\theta_k})}{\partial \hat{\boldsymbol{X}}_i} \boldsymbol{S}_i + \frac{\partial \boldsymbol{\mathcal{F}}_i(\hat{\boldsymbol{X}}(t), \boldsymbol{\theta_k})}{\partial \boldsymbol{\theta}_k}$$

    **else if** *Adjoint SA* **then**
        Integrate $d_t \boldsymbol{\lambda}^T$ backwards in time (store forward solve);

$$d_t \boldsymbol{\lambda}^T = \partial_{\hat{\boldsymbol{X}}} \Psi - \boldsymbol{\lambda}^T \frac{\partial \boldsymbol{\mathcal{F}}}{\partial \hat{\boldsymbol{X}}}, \qquad \boldsymbol{\lambda}(t_f) = 0$$

    **end**

    Obtain predicted probability distribution $\hat{G} = KDE(\hat{\boldsymbol{z}})$

    Compute $\nabla L$ (depends on SA method)

    update $\boldsymbol{\theta}$ using optimizer;
        $\boldsymbol{\theta}_{k+1} = Opt(\boldsymbol{\theta}_k)$
**end**
**Result:** Estimates $\hat{\boldsymbol{\theta}}$ so that Lagrangian model is fitted to SPH data with locally minimized loss

---

## C Additional Results

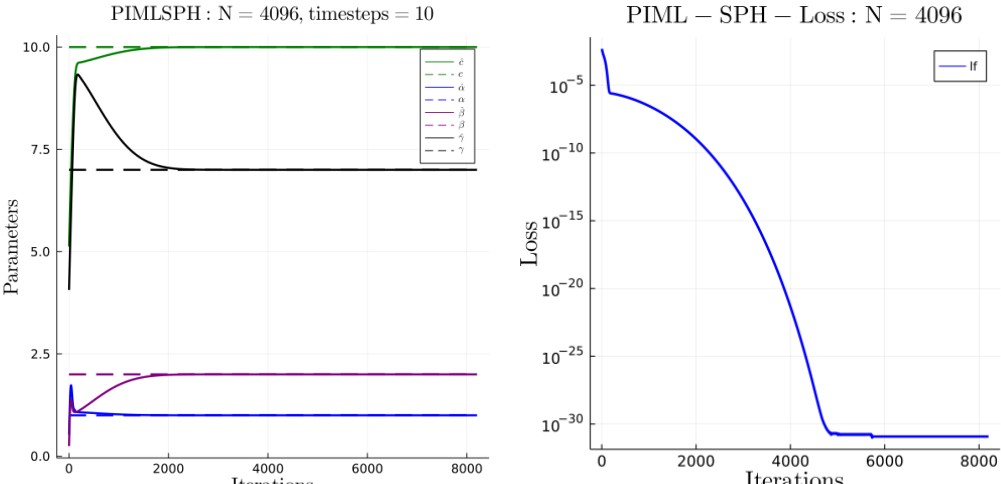

Figure 6: Solving an inverse problem for the fully physics informed model on 3D SPH flow with deterministic external forcing on 4096 particles over 4 SPH parameters. The solid lines show the SPH model parameters converging to the dashed lines representing the ground truth parameters. Here the field based loss function (see subsection 4.4) is used and is converging up to the order of machine precision (as Float64 and the MSE of field difference was used).

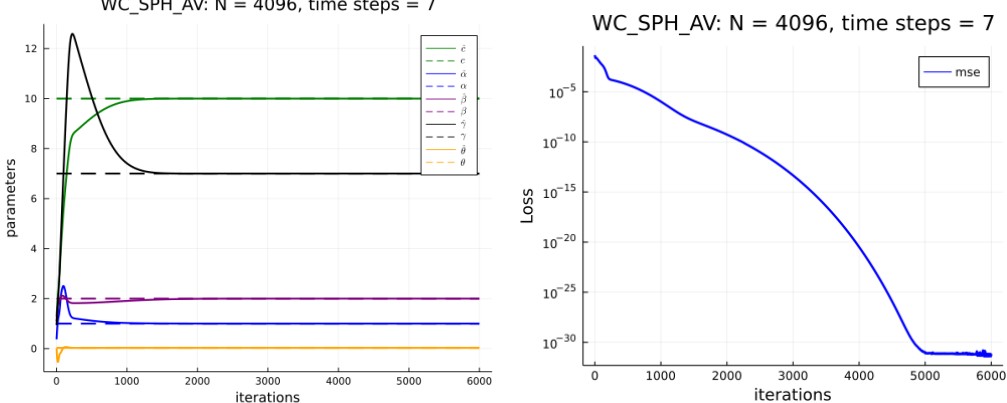

Figure 7: Solving the inverse problems for 3D SPH flow with deterministic external forcing on 4096 particles over 5 parameters (including the rate of energy injection). These plots shows the ability of the mixed mode method to learn the physical parameters of the Physics Informed model Equation 7. The solid lines show the SPH parameters (initially chosen to be uniformly distributed about (0,1)) converged to the dashed lines representing the ground truth parameters. Here the field based loss function (see subsection 4.4) is used with our mixed mode gradient based learning method (see section 4); we see the field based loss (b, d) converging up to the order of machine precision (as Float64 and the MSE of field was used).

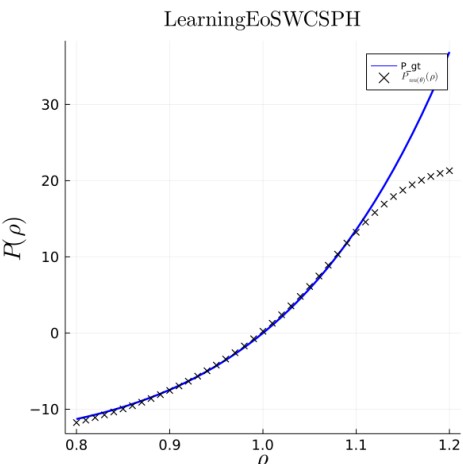

Figure 8: Using a Neural Network to approximate the equation of state (see subsection 4.1) using the $L_{kl} + L_f$ loss function. We see that the Equation of State (EoS) $P(\rho)$ is well approximated on the domain of densities that is seen in training, however, on the Pressure deviates once densities go far beyond range seen in training. Here $P_{gt}$ is the ground truth EoS and $P_{nn}$ is the neural network approximation of the EoS. We note that the underlying ground truth data has density variations within 1% of mean density, so the NN sufficiently approximates the EoS over a larger domain of densities as seen in training.

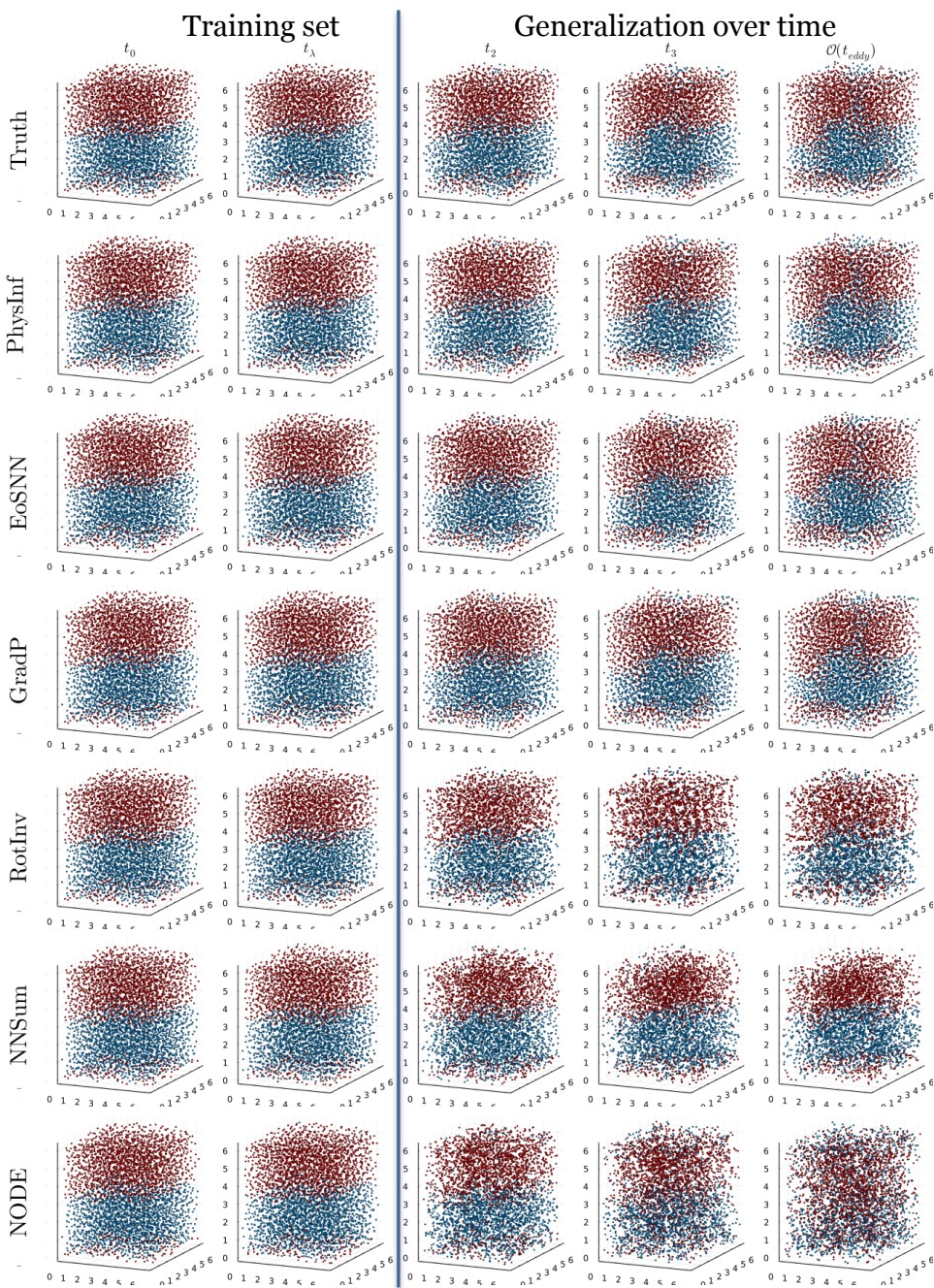

Figure 9: Snapshots of flows comparing the ground truth to the learned models, where learning is done with the $L_{KL} + Lf$ loss function. NODE represents the least informed and Phys Inf is the most informed model. The main takeaway here is that even though the training is occurring on a shortest (physically relevant) time scale $t_\lambda$ (which corresponds to the time of neighboring particles to double their relative separation) the more physics informed models are able to generalize to much longer time scales – all the way to the longest (physically relevant) time scale, $t_{eddy}$ (which is the turnover time scale of the largest eddy of the flow).

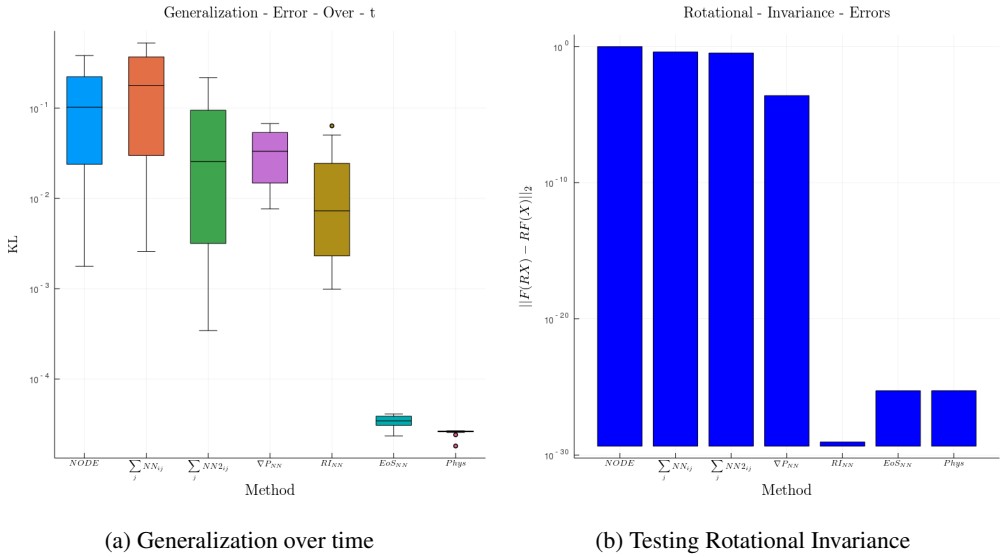

(a) Generalization over time

(b) Testing Rotational Invariance

Figure 10: 2D generalizability where the models are trained and tested using the statistical based loss function. We also see that the rotational invariance (and therefore conservation of angular momentum) is preserved as more physical struture is included in the model

