# OpenReview forum: "Physics Informed Machine Learning of SPH: Machine Learning Lagrangian Turbulence"
_ICLR.cc/2022/Conference — ICLR 2022 Submitted_

### Official Review · Reviewer_sFNH · 2021-10-30

**Correctness:** 2
**Technical Novelty And Significance:** 2
**Empirical Novelty And Significance:** 2
**Recommendation:** 3
**Confidence:** 4

**Main Review:**

The paper claims two contributions: (1) presenting a new physics-informed learning method, i.e. the models in 4.1, mixed SA+AD differentiation, combining MSE+Statistics-based loss, and (2) analyzing these methods, i.e. solving "inverse problems" and the finding that adding physics knowledge increases accuracy and generalization.

Of the models in 4.1, the first one is Neural ODE with a per-node MLP, and the last is the actual PDE that is being learned, with 4 learnable system parameters. The models in-between successively add parts from the PDE into the model (summation, viscosity, kernel, etc.). The most interesting model version is the rotationally equivariant NN (though not exactly novel, given the large amount of papers on SE(3) equivariance in the last few years)
Injecting physical knowledge into learned models is an interesting topic to research, but I was a bit puzzled by this particular approach.
The premise generally is that either we don't know the exact model equation, we have partial or noisy observations, or that ML components can give us other advantages over directly solving the PDE (i.e. faster inference, larger time-steps, differentiability, etc.). Otherwise there seems little point in attempting to use ML in the first place. Neither of these seem to be studied here-- the methods are trained and tested on the exact same PDE that is hard-coded as "physical knowledge", not even noise is added to the data. It is completely unsurprising that the methods which are closest to the PDE that generated the data perform and generalize best; and that the model which have the least learnable components train faster. Beyond this finding, there didn't seem to be any recommendations or pros/cons for the different methods in 4.1.

It's also worth pointing out that there are quite a few papers which do learn from SPH simulation, and already study a wide range of prior knowledge, from strong (the cited Ladicky et al.) to weaker (the cited Ummenhofer et al. and many GNN-based methods, such as Sanchez et al. "Learning complex physics using Graphnets"). It would be a good idea to study this field in context of these established methods as well, and ideally perform comparisons.

The "inverse problem" solved by the fully-physics-informed method is really just the equivalent of training the model-- i.e. fitting the 4 scalar models parameter to the data.

I'm not sure I fully understood the implication of mixing SA and AD, and I'm not sure why this is necessary in the first place; the 'NODE' model is quite literally Neural ODE, so training the model in the same way as Neural ODE would be the obvious choice. Or, alternatively, directly supervising on dri/dt, dvi/dt. (the hard-coded model part should be trivial to implement in a differentiable manner).

The statistics-based loss is an interesting addition-- unfortunately the paper didn't study its effect, or explain why this is necessary when other methods can learn SPH simulations with a plain MSE loss.

**Summary Of The Paper:**

The paper studies several methods for predicting SPH simulations of forced turbulence, in a Neural-ODE style setup. The methods range from general (MLP) to physics-based (i.e. the actual underlying PDE, with a few learnable parameters).
The main finding is that the more physics knowledge is provided, the more accurate and data efficient the learned model becomes.

**Summary Of The Review:**

I don't think the paper in its current form contributes much new insight for the ML community. There isn't much technical innovation in the presented model, and the main finding is quite obvious-- under the conditions studied there really isn't much of a case for using ML methods at all.

---

### Official Review · Reviewer_hB3G · 2021-11-02

**Correctness:** 4
**Technical Novelty And Significance:** 2
**Empirical Novelty And Significance:** 2
**Recommendation:** 6
**Confidence:** 3

**Main Review:**

Strengths:

- The identified problem lies within a very active area of research which is of huge importance in industry.

- The paper is presented in a clear, logical manner with very good supporting supplementary material.

- The methodology is valid: NNs are suitable for regression problems; automatic differentiation and adjoint-based sensitivity analysis are both used for gradient-based optimisation; the proposed loss functions are appropriate.

Weaknesses:

- The most-physics-informed model in the hierarchy is not NN-based, but has a few physically-interpretable parameters to be inferred, which can be done with standard gradient-based optimisation techniques. What are the benefits of using the less-informed, NN-based models over this one? Will the NN-based models perform better on real data? If not, it is difficult to argue their use case.

- It would be worth discussing the relative strengths and weaknesses of the various models, for example by weighing the relative importance of model flexibility offered by using NNs versus model accuracy (or generalization/other metric).

- The quality of the figures could be improved: the axes labels, tick labels and legends are too small to read in every figure, making it difficult to interpret results.

- The sections describing forward, adjoint-based and mixed-mode sensitivity analysis do not add much insight to the paper (as they are not novel, according to the authors), nor are the efficiency benefits over standard automatic differentiation in these models evaluated. Would it not suffice to mention that forward and adjoint-based sensitivity analysis were used to increase the efficiency of gradient calculations (and include details in the appendices as necessary)?

- The MSE loss is introduced but does not appear to have been used. Consider removing it.


**Summary Of The Paper:**

The authors propose a hierarchy of physics-informed, mostly NN-based models of turbulent fluid flow. The NNs are trained to learn various unknown quantities (accelerations or pressures) of the turbulent system’s governing equations (which are derived in the smoothed particle hydrodynamics (SPH) framework) from a synthetic dataset, generated with a fully physics-informed model with known parameters. Automatic differentiation and adjoint-based sensitivity analysis are used to calculate the gradients of the loss functions efficiently depending on the relative sizes of the input and output spaces. The authors compare each model in the hierarchy in terms of accuracy, generalization, interpretability and data efficiency, and show that the more a model is physics-informed, the better it performs on these benchmarks.

**Summary Of The Review:**

Although the paper is technically sound and the results are correct, the use of neural network-based models is not well motivated, given the fully physics-informed model (which outperforms all of the NN-based models) of the system does not require NNs (only standard gradient descent) to find its parameters. Comparing the relative strengths and weaknesses of the different models, and arguing the use-case for the NN-based models would greatly improve the paper. The paper demonstrates the methodology on synthetic data but would be more convincing with an example of it applied to real data.

---

### Official Review · Reviewer_Xkyx · 2021-11-02

**Correctness:** 3
**Technical Novelty And Significance:** 2
**Empirical Novelty And Significance:** 2
**Recommendation:** 3
**Confidence:** 4

**Main Review:**

I’d like to thank the authors for a well composed article. It studies an important direction of applying machine learning to scientific computation and does so in a systematic manner. The main decision is based on strengths and weaknesses expanded below.

Strengths:
* Authors consider a wide range of models;
* Authors consider a relevant set of generalization properties (longer time, different Re)
* Authors provide detailed account of the methods employed to carry out the research + source code

Weaknesses:
* The main conclusion of the paper which is that including additional physics structure improves performance is assessed in a relatively simple setting and rather expected
* The generalization tests do not fully expose limitations of the learned models in the SPH approach, making it hard to assess what big challenges remain/solved using this formulation.
* The use of additional loss functions are of exploratory nature.
* Limited assessment of strengths and weaknesses of the learned Lagrangian vs learned Eulerian approaches.
Systematic comparison of learned simulators is a very challenging task, as the phase space is extremely large. In the present work the models are mostly compared based on loss functions derived for training, as well as statistical distribution of the velocity increments. The losses compare expectedly, with models that are computationally closer to the reference (ground truth simulator) performing better. A better understanding of the structural differences between the errors made by different models could increase the significance of this result.

Generalization tests did not go into details of how models fail under the distribution shift. What is a particular challenge that less structured models struggle with? At which point/type of distribution shift even more structured models start to fall flat?

The additional (to my understanding novel) loss function, such as field reconstruction loss are interesting, but their use was not scrutinized.

Finally the last important point brought up by the paper is the use of learned SPH methods, which is comparatively underrepresented in the literature of learned fluid dynamics models. To generate more interest in this approach a performance comparison between two methods or a clear scoping of what are the central challenges well addressed by the SPH method would be valuable.


**Summary Of The Paper:**

The paper develops a learned fluid dynamics simulator based on smoothed particle hydrodynamics approach. Authors consider a range of models that parameterize the underlying equations of motion for individual degrees of freedom at different levels of granularity (originating from the variation in the amount of physics priors included in the model). A mixed mode AD is used to perform gradient-based optimization of model parameters to minimize discrepancy between reference ground truth data. Authors use a combination of three loss components (particle trajectory, field reconstruction consistency and statistical similarity).

Authors train and evaluate models on a compressible 3D turbulent flow with external forcing. A range of simulation parameters is considered during evaluation, assessing the ability of all models to produce accurate predictions over longer time horizons, as well as simulating more complex flows (different Reynolds numbers). The generalization performance is assessed based on the sum of field reconstruction error and statistical consistency loss. Additional comparison is provided on symmetry properties of learned models, as well as visual comparison of model predictions when trained on very short time sequence;


**Summary Of The Review:**

While the paper reads well and presents results carefully, I found it hard to formulate a clear take away from the paper. This aspect is reflected in the main review section and motivates my rating of "not good enough".

---

### Official Review · Reviewer_mpMw · 2021-11-02

**Correctness:** 4
**Technical Novelty And Significance:** 2
**Empirical Novelty And Significance:** 3
**Recommendation:** 3
**Confidence:** 3

**Main Review:**

Overall, the paper is well written, providing a very good introduction and overview of the modeling tasks. However, I have some concerns regarding the novelty of the work (wrt the methods), that are reflected in my score

strengths:
- a good overview of the problem and motivation of the study
- SA autodiff
- reproducible

weaknesses:
- the contributions seem minimal
- lack discussion and evaluation on some aspects of the considered approaches wrt. efficiency, scalability, robustness to noise etc.

Questions and comments:

- Can you provide more the details about the fully-physics-informed model, and how are the parameters estimated? Are the same parameters used for each particle when generating the data? How would variances/noise in these parameters affect the performance of the models (I don't expect much change in the fully-physical model, but I'm curious about the other)

- It seems that adding the artificial viscosity $\Pi$ improves the generalizability of the models. How was it obtained, were the ground-truth values of $\alpha, \beta$, and $c$ used?

- What is the compute time for each of the NN-based models and how well they scale on problems with more particles and longer trajectories? In this context, in the case of NODEs, how much is m, and how this affects the performance (larger vs smaller neighborhoods)? How much was l?

- Are the same models reported in Fig4 used in the generalizability analysis? What is the performance of the models wrt (combinations of) different loss functions? It seems that the trajectory loss wasn't evaluated, is there a reason for this (other than potential over-fitting)?

- As stated, the mode of the mixed-mode autodiff is chosen a priori for each model separately. But it seems that in all the experiments only FSA was used? How does ASA compare in such cases wrt efficiency?

- Can you elaborate more on the peaks in the loss in Fig.3?

minor:
- Can you elaborate more on the relation between the proposed PIML models and the ones outlined in the related work, that also relay on prior knowledge but attempt at learning parameters (eg. de Anda-Suarez et al 2018)

- I would suggest moving the 'Why PIML challenges' paragraph from the Introduction to the discussion at the end (maybe?). While I enjoyed reading it, it's placement seems to break a bit the flow.

**Summary Of The Paper:**

The paper explores and discusses the effects of incorporating prior domain-knowledge for modeling fluid dynamics with neural networks. The authors focus on smoothed particle hydrodynamics (SPH), a Lagrangian approach for modeling flows, and investigate different alternatives for incorporating physics-based inductive biases when constructing the models. To this end, the paper presents five different alternatives of such models that model particle velocities, with varying levels of prior knowledge: from fully data-driven Neural ODEs to fully physics-informed models with known structure (but unknown physical parameters). The authors also provide details on training such models using sensitivity analysis-based forward and adjoint methods, that are used interchangeably w.r.t the type of the model. Several experiments on synthetic data, in both transductive and inductive setting, show that model-variants that rely more on prior domain-knowledge are more accurate and capable of conserving quantities such as linear and angular momentum.

**Summary Of The Review:**

Overall, the paper is well written, providing a very good introduction and overview of the modeling tasks, which further motivate the importance of using prior knowledge in such scenarios. That said, while the study seems correctly executed, the overall technical contributions of the work seem marginally significant. In particular, in a more general sense, the benefits of knowing particular characteristics of the dynamics (or the whole structure) and using them for modeling dynamics are known. However, in many real-world scenarios this is not the case, which leads to different approaches for data-driven modeling. The authors set-out to evaluate this trade-off (in a synthetic setting), but do not provide deeper discussion between the model-variants across different aspects (such as efficiency, accuracy, memory, scalability, robustness (to noise) etc) which might be of more interest for the ICLR audience.

---

### Official Review · Reviewer_jmEb · 2021-11-03

**Correctness:** 3
**Technical Novelty And Significance:** 2
**Empirical Novelty And Significance:** 3
**Recommendation:** 5
**Confidence:** 3

**Main Review:**

Based on the smoothed particle hydrodynamics (SPH) method, this paper proposed and explored a fair amount of models with different levels of prior knowledge embedded.
When evaluating the learned results, it's good that this paper also considers different kinds of physical properties and the model's generalization ability, other than just comparing the trajectory.

However, in the experiment parts, there should be some comparisons with other existing learning-based fluid simulators, especially the methods that also use the Lagrangian view, to show how well the proposed physics-informed model performs compared with others. For better comparison, it's also helpful to show some results of the training and inference time used by different models.



**Summary Of The Paper:**

This paper introduces a hierarchy of parameterized models based on one of the Lagrangian simulation methods, Smoothed particle hydrodynamics (SPH).
Several models with different levels of physics information embedded are proposed.
When training these models, sensitivity analysis and automatic differentiation are used to compute the gradients, and different loss functions (trajectory-based, field-based, and KL divergence) are used.
In the experiments, this paper evaluates the ability of the parameterized SPH models to learn the system's properties from data and to generalize to different environments.


**Summary Of The Review:**

Although a certain amount of experiments is provided in this paper, it lacks some comparison results to show how well the proposed model performs compared with other existing methods.
Adding SPH-related physics information to the model architectures can be a good direction to improve the data-driven simulator, but the models used in this paper seem to be simply replacing some parts in the classical SPH solver, which makes me have some concern about the novelty.

---

### Decision · Program_Chairs · 2022-01-20

**Decision:**

Reject

**Comment:**

The paper explores and discusses the effects of incorporating prior domain-knowledge for modeling fluid dynamics with neural networks, with a focus on smoothed particle hydrodynamics. Reviewers agree that the contributions are modest, and that they are not well presented with respect to issues of efficiency, scalability, robustness, etc.  More work need to be done to make it useful to the community.  Many of directions to improve the paper were in reviewer comments.